# Induction of Extrinsic Apoptotic Pathway in Pancreatic Cancer Cells by *Apteranthes europaea* Root Extract

**DOI:** 10.3390/ijms262010221

**Published:** 2025-10-21

**Authors:** Rinat Bar-Shalom, Lana Abdelhak, Wafa Zennouhi, Farid Khallouki, Laila Benbacer, Fuad Fares

**Affiliations:** 1Department of Human Biology, Faculty of Natural Sciences, University of Haifa, Mount Carmel, Haifa 3498838, Israel; rbar-shal@staff.haifa.ac.il (R.B.-S.); lanabdelhak8@gmail.com (L.A.); 2Team of Ethnopharmacology and Pharmacognosy, Biology Department, Faculty of Sciences and Techniques, Moulay Ismail University of Meknes, 52000 Errachidia, Morocco; w.zennouhi@edu.umi.ac.ma (W.Z.); f.khallouki@fste.umi.ac.ma (F.K.); 3Biology Unit and Molecular Research, Department of Life Sciences, National Center for Energy, Sciences, and Nuclear Techniques (CNESTEN), P.O. Box 1382, 10001 Rabat, Morocco; lbenbacer@yahoo.com

**Keywords:** phytochemicals, natural products, cytotoxicity, apoptosis induction, caspases, pancreatic cancer therapy

## Abstract

Pancreatic cancer is an extremely deadly disease with few effective treatment options and the lowest survival rate among all types of cancer. As a result, there is an urgent need for the development of new and more effective treatment strategies. Natural products have long been a vital source of drug discovery, offering unique bioactive compounds, and representing a promising source for new, effective, and less toxic treatments. In the present study, we aimed to investigate the effects of *Apteranthes europaea* (Guss.) Murb (*A. europaea*) root extract on the growth of pancreatic cancer cells. The proliferation assay (XTT) and real-time analysis using the IncuCyte Live-Cell Analysis System, following treatment of PL45 and Mia PaCa-2 pancreatic cancer cells with escalating concentrations (50–200 µL) to *A. europaea* root extract, demonstrated the progression of apoptosis. Apoptosis induction was confirmed through cell cycle analysis and Annexin V/PI double staining assays. Western blot analysis revealed the distinct activation of caspase-8, accompanied by the cleavage of caspase-3 and Poly (ADP-ribose) polymerase (PARP). Interestingly, no activation of caspase-9 was observed, suggesting the involvement of the extrinsic apoptotic pathway. Our findings suggest that *A. europaea* extract may be a potential novel strategy for treating pancreatic cancer.

## 1. Introduction

Pancreatic ductal adenocarcinoma (PDAC) is one of the deadliest cancers and is expected to become the second-leading cause of cancer-related deaths in the United States by 2030 [1]. The aggressive nature of PDAC is characterized by early metastasis, late-stage diagnosis, and profound resistance to conventional therapies. Despite advancements in systemic therapies, such as surgery, radiation, immunotherapy, and targeted treatments, improvements in survival rates have been modest, with only approximately 12% survive beyond five years [2]. This poor prognosis is largely attributed to the fact that the majority of patients are diagnosed at an advanced, inoperable stage, when therapeutic options are severely limited. Conventional chemotherapy regimens currently used for PDAC include gemcitabine (often in combination with nab-paclitaxel) and FOLFIRINOX (a combination of 5-fluorouracil, leucovorin, irinotecan, and oxaliplatin). For example, the combination of gemcitabine and nab-paclitaxel has shown improved response and overall survival compared with gemcitabine alone in locally advanced disease [3]. While these treatments can extend survival, they are associated with significant side effects including severe neutropenia, peripheral neuropathy, gastrointestinal toxicity (nausea, vomiting, diarrhea), fatigue, and in the case of FOLFIRINOX, increased risks of febrile neutropenia and thrombocytopenia [4,5]. Therefore, the development of new, safe, and effective therapeutic agents remains a critical priority in the fight against this malignancy.

Natural products have played a significant role in the history of anticancer drug development [6]. The remarkable chemical diversity found in nature has provided a vast array of bioactive compounds with significant therapeutic potential [7,8,9]. Between 1981 and 2019, it was estimated that approximately 25% of all newly approved anticancer drugs were derived from or related to natural products [10], highlighting the continued importance of natural product research in oncology. Plant-derived compounds offer several advantages, including structural diversity, potential for mechanisms of action, and the potential to overcome drug resistance through novel molecular targets [11]. Different parts of plants, including seeds, roots, leaves, fruits, flowers, or even entire plants, have long been utilized for medicinal purposes for their medicinal properties [12]. Medicinal plants are particularly valued for their safety, effectiveness, affordability, and easy accessibility [13], making them attractive candidates for drug discovery and development.

The medicinal plant, *Apteranthes europaea (A. europaea*) is a species of the *Apteranthes* genus, a member of the Apocynaceae family. *A. europaea* is a low growing, perennial, mat-forming succulent plant (5–25 cm in height) [14]. It is widely distributed across Mediterranean countries including Morocco, Algeria, Egypt, Spain, Tunisia, and Italy [15]. In traditional medicines, particularly in Morocco, *A. europaea* is used as a remedy plant that exhibits several medical uses such as for diabetes, digestive problems, anti-inflammation, bone disorders, cancer, and reproductive system diseases [16]. Several research studies have confirmed the antimicrobial, anti-inflammatory, antifungal [17], antioxidant [18], and hepatoprotective effects of *A. europaea* [19]. Moreover, *A. europaea* extracts showed antitumor activity against human leukemia, liver [20], breast [18], prostate, and colorectal cancer [21]. A recently published study analyzed the hydroethanolic extract of the aerial parts of *A. europaea*, identifying several bioactive compounds, such as phenolic acids (e.g., ferulic acid and vanillic acid) and flavonoids (e.g., naringenin, quercetin, and myricetin). These compounds are known for their antioxidant, anti-inflammatory, and anticancer properties, which likely contribute to the hepatoprotective and antitumor effects observed in experimental studies. Additionally, this study demonstrated that the hydroethanolic extract from *A. europaea* aerial parts, enriched with flavonoids, polyphenols, and saponin, effectively reduced the expression of pancreatic cancer-associated markers [22].

To the best of our knowledge, no scientific evidence has been published regarding the apoptotic mechanism induced by *A. europaea* root extract on pancreatic cancer cells. This critical gap in knowledge highlights the necessity for further research, particularly in PDAC. This study aimed to examine the anticancer effects of *A. europaea* extract in vitro and to determine the molecular mechanism underlying the tumor growth inhibition induced by the extract in PDAC. Investigating these mechanisms may provide new insights into the therapeutic potential of *A. europaea* and pave the way for development of novel, plant-based treatments for pancreatic cancer.

## 2. Results

### 2.1. A. europaea Extract Inhibits the Proliferation of Human Pancreatic Cancer Cells

A viability assay was performed on PL45 as well as Mia PaCa-2 cells treated with *A. europaea* root extract administered in a concentration range 50–200 µg/mL for 24 , 48,  or 72 h as reported in Figure 1. The results indicated that the extract significantly (*p* < 0.001) decreased cell viability in both cell lines in a time- and dose-dependent manner (Figure 1A). According to a two-way ANOVA test, a strong correlation was found between the time of treatment and concentration in both cell lines (PL45; F(8) = 15.170, R^2^ = 0.808, *p* < 0.001. MIA-PaCa; F(7) = 2.895, R^2^ = 0.889, *p* = 0.006). Cell viability, as measured for PL45 (Figure 1A) was decreased by 44.8 ± 11.36 and 59.62 ± 11.77 (*p* < 0.001) following 72 h of treatment with 150 and 175 µg/mL of *A. europaea* extract, respectively. Cell viability of Mia PaCa-2 (Figure 1B) decreased by 57.7 ± 3.88 and 82.9 ± 2.39 (*p* < 0.001) following 48 h of treatment with 150 and 175 µg/mL of the extract, respectively.

To assess whether *A. europaea* extract treatments induced excessive membrane damage and necrotic cell death, a lactate dehydrogenase (LDH) leakage assay was performed. PL45 and Mia PaCa-2 cells were treated with increasing doses (50–200 µg/mL) of the extract for 24 h. It was observed that the extract, at the tested concentrations, does not cause a statistically significant change in LDH level in the media, compared with untreated control cells (Appendix A).

### 2.2. The Effect of A. europaea Extract on Cell Cycle Progression

PL45 and Mia PaCa-2 cells treated with *A. europaea* extract were collected, stained with PI, and analyzed for cell cycle distribution using a flow cytometer (Figure 2). The results revealed a significant increase (one-way ANOVA; *p* < 0.01) in the sub-G1 phase population of PL45 cells, rising from 5.06 ± 1.24% to 79.4 ± 8.3% within 72 h of treatment with 175 µg/mL of the extract (Figure 2(a2)). Concurrently, the proportions of cells in the S and G2/M phases decreased from 11.2 ± 1.3 to 1.8 ± 0.67% and 27.66 ± 0.72 to 4.23 ± 1.82%, respectively. Statistical analysis using two-way ANOVA indicated a significant interaction between treatment duration, extract concentration, and cell cycle phase (F(9) = 26.260, *p* < 0.001). Additionally, a separate two-way ANOVA comparing treatment duration and concentration at the sub-G1 phase demonstrated a strong correlation between these factors (F(3) = 25.169, *p* < 0.001). Exposure of Mia PaCa-2 cells to 150 µg/mL and 175 µg/mL (Figure 2b) of the extract for 36 h resulted in sub-G1 populations of 38.67 ± 3.88 and 55.5 ± 4.37, respectively. A one-way ANOVA with post hoc comparisons revealed a significant increase in sub-G1 populations at both concentrations compared with the control (untreated cells) (F (2,9) = 39.235, R^2^ = 0.874, *p* < 0.001).

### 2.3. A. europaea Extract Induces Apoptotic Cell Death in Pancreatic Cancer Cells

Annexin V-FITC and propidium iodide (PI) staining were used to quantify the proportion of cells undergoing apoptosis and necrosis and those remaining viable. Apoptotic cells were counted as early apoptotic cells (quadrant Q2) and late apoptotic cells (quadrant Q4) and represented as percentages of apoptotic cells (Figure 3b,d) from the total cell population. Figure 3a,b indicate the effect of treating PL45 cells with 125, 150, and 175 µg/mL of *A. europaea* extract for 48 and 72 h. In the control group, the apoptotic rate was 9.95 ± 1.02% at 48 h and 9.61 ± 0.9% at 72 h (Q2 + Q4). Treatment with 175 µg/mL of the extract significantly increased apoptotic rates to 29.7 ± 6.12% after 48 h (*p* < 0.01) and 51.82 ± 6.46% after 72 h (*p* < 0.01). Additionally, Annexin V increased by 1.67, 2.5 (*p* < 0.05), and 5.4 (*p* < 0.01) folds following treatment with 125, 150, and 175 µg/mL, respectively, compared with the control. Similarly, Mia PaCa-2 cells (Figure 3c,d) treated with 150 and 175 µg/mL for 36 h exhibited a significant increase (*p* < 0.01) in Annexin V by 2.54 and 2.83 fold, respectively, relative to untreated cells. These results indicate that *A. europaea* extract induces apoptotic cell death in pancreatic cancer cells.

To strengthen the results obtained, the apoptotic effect of *A. europaea* extract was further validated using real-time analysis using the IncuCyte live-cell imaging system in PL45 and Mia PaCa-2 pancreatic cell lines. Cells were treated with increasing concentrations of *A. europaea* extract (0, 125, 150, and 175 µg/mL) and monitored over time using Annexin V-FITC and PI staining. Four images per well were captured every 2 h over a period of 72 h for PL-45 cells and 36 h for Mia PaCa-2 cells (Figure 4). Quantitative analysis revealed a dose- and time-dependent increase in apoptosis in both cell lines, as indicated by the accumulation of Annexin V-positive (green) cells. In PL45 cells, treatment with 150 and 175 µg/mL induced a marked apoptotic response evident from 24 h and increasing progressively through 72 h. According to post hoc statistical testing, no significant difference was observed between these two concentrations. Treatment with 125 µg/mL led to moderate increases in apoptosis compared with the untreated control, which exhibited minimal Annexin V staining (Figure 4(a2)). Mia PaCa-2 cells showed a similar trend (Figure 4(b2)). The highest levels of apoptosis were observed following treatment with 150 and 175 µg/mL between 24 and 36 h, with no statistically significant difference between these two concentrations. A less pronounced apoptotic effect was observed at 125 µg/mL. Fluorescence imaging further supported these findings, showing a clear increase in Annexin V and PI staining over time and with increasing extract concentration, consistent with early and late stages of apoptosis, respectively. Morphological changes characteristic of apoptosis were also noted, with treated cells appearing smaller and more condensed compared with untreated cells. Two-way ANOVA test confirmed a statistically significant interaction between treatment concentration and time in both cell lines. In PL45 cells, the interaction was significant at timepoints 0, 24, 48, and 72 h (F(9) = 64.745, *p* < 0.001, R^2^ = 0.947). In Mia PaCa-2 cells, a significant interaction was also observed at 0, 24, and 36 h (F(11) = 140.533, *p* < 0.001, R^2^ = 0.921). These findings confirm that *A. europaea* extract induces apoptosis in a dose- and time-dependent manner.

### 2.4. A. europaea Extract Activates the Caspase-Mediated Apoptotic Pathway

To investigate the cell death mechanisms induced by *A. europaea* extract, Western blot analysis was performed to examine key apoptotic markers associated with both the extrinsic (caspase-8) and the intrinsic (caspase-9) apoptosis cascade. As shown in Figure 5, treatment with *A. europaea* extract of PL45 and Mia PaCa-2 cells with 175 µg/mL for 48 and 72 h and 150 µg/mL for 36 h, respectively, resulted in the activation of caspase-8 and caspase-3, followed by the cleavage of poly (ADP-ribose) polymerase (PARP). Notably, caspase-9 activation was not activated. These findings may suggest that *A. europaea* extract induces apoptosis in both cell lines through the extrinsic pathway.

## 3. Discussion

In the present study, we demonstrate that the methanolic extract of *A. europaea* roots exerts significant antiproliferative activity on two human pancreatic cancer cell lines, PL45 and Mia PaCa-2, in a dose- and time-dependent manner. According to the XTT assay, cell viability was significantly reduced following exposure to *A. europaea* root extract with 100 and 125 µg/mL for Mia PaCa-2 and PL45 cells, respectively. Furthermore, the reduction in cell viability was not associated with increased levels of LDH release, suggesting that the extract did not induce nonspecific membrane damage or necrosis, but rather elicited regulated cell death. According to Amrati, F.E.Z. et al., [23] the aerial parts of *A. europaea* were tested on Mia PaCa-2 cell line with no reduction in cell survival following exposure to 10, 100, and 1000 µg/mL of the hydroethanolic extract of *A. europaea* following 72 h of treatment [23].

From a phytochemical point of view, *A. europaea* has been well-documented. Studies on different parts of the plant, using solvents like methanol, hydroalcoholic mixtures, and ethyl acetate, have consistently shown it to possess a rich and diverse profile of compounds, primarily phenolics. Chromatographic analyses of various extracts have identified a wide array of flavonoids, including luteolin and its glycosylated derivatives (luteolin-3′,4′-O-diglucoside, luteolin-4′-O-neohesperidoside, luteolin-7-O-glucoside), quercetin (both as aglycone and quercetin-3-O-rutinoside/rutin), kaempferol (and its derivative kaempferol-3-O-hexose deoxyhexose), myricetin, apigenin-4′-O-neohesperidoside, hesperetin, catechin, and epicatechin. The phenolic acid composition is equally broad, encompassing gallic acid, trans-ferulic acid, caffeic acid, chlorogenic acid, syringic acid, salicylic acid, p-coumaric acid, rosmarinic acid, vanillic acid, 3,4-dihydroxybenzoic acid, 2-hydroxycinnamic acid, sinapic acid, and ascorbic acid. This consistent identification of bioactive phenolics across hydroethanolic, ethyl acetate, and methanolic extracts firmly establishes *A. europaea* as a significant source of these compounds, providing a chemical basis for its investigated biological activities [6,14,18,20,22].

Different plant parts contain varying concentrations of bioactive compounds, resulting in distinct anticancer activities. Plant parts such as leaves, stems, flowers, and roots can differ significantly in their polyphenol and flavonoid content, leading to variable therapeutic outcomes. This variation in efficacy has been demonstrated across multiple studies [24,25,26,27]. For instance, hydromethanolic extract from *Bauhinia variegata* floral buds showed superior antitumor activity against melanoma compared with leaf and stem bark extract [24]. Similarly, aqueous leaf extract of *Inula viscosa* demonstrated a significant anticancer effect against colorectal cancer cells in both in vitro and in vivo models [25]. However, this variability can also result in inconsistent outcomes, as demonstrated by Anglana et al. [26], who found that aqueous extracts from aerial parts of *Inula viscosa* exhibited moderate cytotoxicity against SW620 colorectal cancer cells but had no significant effect on DLD-1 and HT-29 cell lines. Moreover, it is important to note that different solvents used for *A. europaea* extractions revealed different molecules present in the extract with dependency on solvents used. This solvent-dependent variability has been well demonstrated in multiple studies. Zazouli et al. [27] investigated root extracts of *A. europaea* using solvents of increasing polarity—hexane, chloroform, dichloromethane, ethyl acetate, acetone, ethanol, and methanol—and showed that methanol extracts exhibited the highest total phenolic content, while ethanol and ethyl acetate were particularly rich in flavonoids. In contrast, non-polar solvents such as hexane and chloroform yielded much lower phenolic content, supporting the idea that solvent polarity plays a key role in determining extract composition. Atrooz et al., [28] suggest that the efficacy of *A. europaea* extracts against MCF7 cells is significantly affected by the type of solvent used. They evaluated various extracts—including chloroform, methanolic, ethyl acetate, and aqueous—and found that the chloroform extract exhibited the most potent antiproliferative activity. Similarly, Amrati et al. [22,23] showed that hydroethanolic, polyphenolic, and n-butanol extracts from the aerial part of *A. europaea* contained distinct bioactive compounds, depending on the solvent used. Among them, the hydroethanolic extract demonstrated the strongest antitumor activity against pancreatic cancer cell lines (Mia PaCa-2 and BxPC-3), highlighting the importance of solvent choice in both phytochemical composition and biological efficacy. Moreover, the aerial parts of *A. europaea* used to extract bioactive compounds with methanolic solvents, revealed polyphenols, flavonoids, and glycosides as the primary constituents. In contrast, in our study we used the root extract and not isolated fractions. Using an extract offers several advantages in therapeutic applications compared with isolated fractions. One of the key benefits is the presence of multiple bioactive compounds that can work synergistically to enhance the extract’s overall efficacy. This synergy often results in broader-spectrum activity, as different compounds may target multiple pathways simultaneously, making the extract effective against complex conditions such as cancer. Additionally, the presence of diverse compounds in the extract can mitigate potential toxic effects that may arise when using a single, highly concentrated constituent [29].

Flow cytometry analysis revealed that *A. europaea* extract induced a significant, time- and dose-dependent accumulation of pancreatic cancer cells in the sub-G1 phase, indicating apoptotic DNA fragmentation. PL45 cells exhibited a notable sub-G1 increase after 72 h at 175 µg/mL, while Mia PaCa-2 cells responded to lower concentrations (150–175 µg/mL) within 36 h. Two-way ANOVA confirmed significant interactions between treatment duration, concentration, and cell cycle phase. IncuCyte live-cell imaging analysis and Annexin V-FITC staining further validated increased early and late apoptotic populations in both cell lines, particularly at higher concentrations and longer exposures. Notably, Mia PaCa-2 cells were more sensitive, exhibiting earlier apoptotic responses. These findings are consistent with previous reports on the methanolic extract of *A. europaea*, which induced apoptosis in a cell line-dependent manner [21]. These findings suggest that *A. europaea* extract induced a concentration-dependent effect in HT-29 but not in HCT116 cells, and PC-3 cells showed a pronounced sub-G1 peak compared with other lines. Such variability highlights the importance of tailoring therapeutic approaches to specific cancer cell types. Similar trends have been reported with other plant extracts, where apoptosis is often both time- and dose-dependent but varies between cell lines. For example, extracts rich in polyphenols or alkaloids often show faster apoptotic responses in highly proliferative cancer cells, while less aggressive cells may require prolonged exposure for comparable effects [30]. This aligns with the rapid apoptotic response seen in Mia PaCa-2 cells and the slower, but ultimately significant, apoptosis observed in PL45 cells.

To elucidate the molecular mechanisms underlying the observed apoptotic effect, we examined the activation status of key caspases and the cleavage of poly (ADP-ribose) polymerase (PARP), a hallmark of apoptosis. Apoptosis is primarily mediated through two main signaling cascades: the extrinsic pathway (death receptor-mediated), activated by initiator caspases such as caspase-8 or caspase-10, and the intrinsic (mitochondrial) pathway, initiated by caspase-9. Both pathways converge on the activation of effector caspases—caspase-3, caspase-6, and caspase-7—which mediate the proteolytic degradation of various intracellular targets, ultimately leading to the regulated fragmentation of the cell [31]. Since many chemotherapeutic agents induce apoptosis, resistance to apoptosis is a key actor in treatment failure in cancer, including PDAC [32]. Our results revealed robust cleavage of caspase-8 and caspase-3, as well as PARP, in both pancreatic cell lines following treatment with the root extract. Notably, no cleavage of caspase-9 was detected, suggesting that the apoptotic pathway activated by the extract primarily involves the extrinsic pathway rather than the intrinsic pathway. This observation is of particular interest, as pancreatic cancer cells often develop resistance to intrinsic apoptotic signaling due to multiple molecular alterations affecting the mitochondrial pathway, such as p53 mutations and the dysregulation of Bcl-2 family proteins [33,34,35]. Given that most chemotherapeutic agents primarliy target the intrisic pathway, the ability of *A. europaea* extract to activate the extrinsic pathway may represent a potential therapeutic advantage in overcoming apoptosis resistance mechanisms commonly observed in pancreatic cancer. Future studies will include qRT-PCR and/or transcriptomic analyses to validate the involvement of specific genes in the pathway we propose.

These findings highlight the potential of targeting the extrinsic apoptotic pathway as an alternative or complementary strategy for overcoming apoptosis resistance in PDAC. Importantly, beyond its pro-apoptotic effect, *A. europaea* hydroethanolic aerial parts’ extract and its bioactive constituents—including flavonoids, polyphenol-rich fractions, and saponins—were also found to downregulate key markers of cancer stemness, such as Oct-4 and Nanog proteins, as well as CD133 and Sox2 mRNA, in a dose-dependent manner in pancreatic cancer cell lines [22]. These stemness-associated factors are strongly implicated in chemoresistance and tumor relapse. Consistent with this, Amrati et al., [23] have previously demonstrated that *A. europaea* extracts sensitize pancreatic cancer cells to chemotherapy. Together with our results, these findings underscore the dual therapeutic potential of *A. europaea* extract—not only in promoting apoptosis through the extrinsic pathway but also in impairing cancer stem-like properties, thereby enhancing chemosensitivity and potentially reducing recurrence in PDAC.

Previous studies on *A. europaea* have reported anticancer activity in various cancer types, including breast [18,28], liver [20], leukemia [20,36], prostate, and colon cancers [29], often accompanied by apoptosis induction and rarely by caspase activation. Notably, Samiry et al. [29] described the activation of caspase-3 and cleavage of PARP in colon and prostate cancer cells following exposure to methanolic extract from the aerial parts of *A. europaea*. However, most of these studies either focused on aerial parts of the plant or lacked detailed mechanistic insights. In contrast, our study contributes novel findings by demonstrating, for the first time to our knowledge, that *A. europaea* root extract activates the extrinsic apoptotic pathway in pancreatic cancer cells. This positions the extract as a promising candidate for further exploration in the context of PDAC treatment, particularly given the urgent need for agents capable of bypassing resistance mechanisms associated with the mitochondrial pathway.

In conclusion, our findings demonstrate that the root extract of *A. europaea* exerts a strong anticancer effect against pancreatic cancer cells, primarily through the selective activation of the extrinsic apoptotic pathway. This specific mechanism of action is particularly significant considering the well-known resistance of pancreatic cancer to most conventional chemotherapeutic agents. These results not only broaden the pharmacological understanding of *A. europaea* but also emphasize the potential of underexplored medicinal plants as promising reservoirs of novel anticancer compounds. However, several limitations of the present study should be acknowledged. The data were obtained exclusively from in vitro experiments, which do not fully reproduce the complex biological environment of living systems. The interpretation of these findings is constrained by the absence of in vivo validation, the limited exploration of molecular mechanisms, and the lack of isolation and structural identification of the active constituents. Moreover, essential pharmacological parameters such as the dose–response behavior, pharmacokinetic properties, and long-term safety of the extract remain to be determined. Future investigations should therefore aim to confirm these anticancer effects in relevant animal models of pancreatic cancer, conduct bio-guided fractionation to help isolate and characterize the most active molecules, and apply advanced analytical (UPLC–MS/MS, GC–MS) and molecular techniques to clarify the precise signaling pathways involved. It is also important to consider that the composition and concentration of secondary metabolites in *A. europaea* may differ from previously reported data due to multiple factors, including plant age, root size, environmental and habitat conditions, as well as post-harvest processing and extraction methods. Addressing all these aspects will be crucial to help evaluate the clinical relevance and translational potential of *A. europaea* as a candidate for anticancer drug development. Overall, this study establishes a solid piece of scientific basis for the continued exploration of *A. europaea* as a promising natural source of innovative therapeutic agents targeting one of the most aggressive and treatment-resistant malignancies worldwide.

## 4. Materials and Methods

### 4.1. Preparations of the Plant Extracts

*Apteranthes europaea* was collected in June 2023 from the Zaouiat Aoufous region, southeastern Morocco (31°42′ N, 4°9′ W). A voucher specimen RCE2023, was deposited at the FSTE, Errachidia. The small root parts of the plant were washed with distilled water, shade-dried, and ground into a fine powder using a mechanical grinder. A successive solvent extraction was performed using a Soxhlet apparatus with solvents of increasing polarity: cyclohexane, chloroform, ethyl acetate, and methanol. For each step, 10 g of the powdered root material was extracted with 100 mL of solvent until exhaustion, indicated by the solvent running clear. The methanolic extract (7% yield), selected for biological testing, was concentrated under reduced pressure at low temperature. The resulting residue was stored at 4 °C. Prior to use, the extract was dissolved in absolute ethanol and kept at –20 °C until analysis.

### 4.2. Cell Cultures

The human pancreatic cancer cell lines Mia PaCa-2 (poorly differentiated) and PL45 (moderately to well-differentiated) (ATCC, Rockville, MD, USA) were maintained in DMEM medium, supplemented with 1% L-glutamine, 10% fetal bovine serum (FBS), 1% PenStrep (penicillin + streptomycin), and 1% sodium pyruvate (Biological industries, Beit HaEmek, Israel). Cells were grown in a humidified incubator at 37 °C with 5% CO_2_ in the air and served twice a week with fresh medium. The cell lines were routinely tested for mycoplasma contamination with the Mycoplasma Test Kit EZ-PCR. All cell culture reagents, including the Mycoplasma Test Kit, were supplied by Biological Industries (Beit Haemek, Israel).

### 4.3. XTT Cell Proliferation Assay

Evaluation of plant extract effect on cell viability was performed by the XTT assay, which is used to measure cellular metabolic activity as an indicator of cell viability, proliferation, and cytotoxicity. PL45 and Mia PaCa-2 cells at a cell density of (10^4^) were seeded in 150 μL of medium, using 96-well plates. After 24 h, the crude extract was added in several concentrations: 50, 75, 100, 125, 150, 175, and 200 μg/mL for a period of 24, 48, and 72 h. Control wells were medium treated wells. Following the treatment, the cell viability was determined by the XTT assay (Biological Industries, Beit HaEmek, Israel), according to the manufacturer’s instructions, using a plate reader (BioTek, Winooski, VT, USA) at 450 nm wave and subtracted from the reference absorbance at 620 nm. Experiments were repeated 2–5 times independently and conducted in at least 3 replicates. Data were presented as the average proliferation percentage of the respective control.

### 4.4. Cytotoxicity Assay

To distinguish between cytotoxic effects and excessive necrotic cell death following *A. europaea* extract treatments, lactate dehydrogenase (LDH) leakage assay was performed. LDH, a cytoplasmic enzyme, is rapidly released from the cells into the medium when the plasma membrane is damaged. The integrity of the plasma membrane following treatment was determined by measuring the LDH activity in the culture medium. Briefly, PL45 and Mia PaCa-2 cells were cultured in 96-well plates. *A. europaea* extract was added in different concentrations (50–200 μg/mL). Untreated cells served as negative controls. The levels of LDH in the cell culture media were detected by the CyQUANT^TM^ LDH Cytotoxicity Assay (Invitrogen, Waltham, MA, USA) following the manufacturer’s instructions 24 h post treatment. All experiments were conducted in triplicate, and data were presented as the average of three independent experiments (mean ± SE) and expressed as percentages of respective controls.

### 4.5. Cell Cycle Analysis

For cell cycle distribution analysis, PL45 and Mia PaCa-2 cells (10^6^) were treated with *A. europaea* root extract at concentrations of 125, 150, and 175 µg/mL for PL45 and with 150 and 175 µg/mL for Mia PaCa-2 cells, for 48 and 72 h for PL45 and 36 h for Mia PaCa-2 cells. At the end of incubation period, cells were trypsinized, harvested and collected with growth medium, and centrifuged at 2000 rpm for 5 min at 4 °C. Cells were twice washed with cold PBS (Biological Industries, Beit HaEmek, Israel) and then fixed in pre-chilled 70% ethanol at −20 °C for one hour. The cells were incubated with 0.1% NP-40 on ice for 5 min and subsequently washed twice with cold PBS, each time by centrifugation at 2000 rpm for 5 min at 4 °C. Then, 1 mL of cold PBS containing RNase (100 µg/mL) was added to cells for 30 min. Finally, 50 µg/mL of propidium iodide (PI) (Sigma-Aldrich, St. Louis, MO, USA) was added to cells followed by incubation for 20 min on ice. DNA content was examined by flow cytometry using FACSCantoII with BDDiva V9 software (Becton Dickenson, San Jose, CA, USA).

### 4.6. Annexin V/PI Double Staining Assay

Apoptotic cell death was further analyzed using FITC-labeled Annexin V and propidium iodide (PI) with an Annexin V-FITC apoptosis detection kit (MBL, Nagoya, Japan), according to the manufacturer ‘s instructions. Briefly, cells (2 × 10^5^) were seeded in 25 cm^2^ flasks and allowed to attach overnight. PL45 cells were exposed with 125, 150, and 175 μg/mL of *A. europaea* extract for 48 or 72 h, while Mia PaCa-2 cells were treated with 150 and 175 μg/mL for 36 h. To detect early and late apoptosis, both adherent and floating cells were collected together. Treated and untreated cells were harvested by trypsinization, washed, and suspended in ice-cold PBS. The cell pellet was then resuspended in ice-cold binding buffer containing FITC-conjugated Annexin V and PI. The samples were incubated in the dark at room temperature for 15 min before analysis using FACSCantoII Flow cytometry (Becton Dickenson, San Jose, CA, USA). The Annexin V-FITC-negative/PI negative population was classified as normal healthy cells. Annexin V-FITC-positive/PI negative cells were considered early apoptotic. Cells positive for both Annexin V-FITC and PI were classified as late apoptotic, while those negative for Annexin V-FITC but positive for PI were identified as necrotic cells. The percentage distributions of normal, early apoptotic, late apoptotic, and necrotic cells were calculated using BDDiva V9 software (Becton Dickenson, San Jose, CA, USA).

### 4.7. Incucyte Imaging System—Time-Lapse Imaging of Apoptosis Using Annexin V/PI Staining

The apoptotic effect of *A. europaea* extract on PL 45 and Mia PaCa-2 pancreatic cancer cell lines was assessed in real time using the IncuCyte SX5 Live-Cell Analysis System (Sartorius, Bohemia, NY, USA) with Annexin V labeling, as previously described [37]. Mia PaCa-2 and PL-45 cells were seeded at a density of 1.2 × 10^4^ cells/well into 96-well plate and allowed to adhere overnight. Cells were then treated with three concentrations of the *A. europaea* extract (125, 150, and 175 µg/mL) or vehicle control, with each condition performed in triplicate. Cells were stained with Annexin V conjugated to a green fluorescent probe and propidium iodide (PI). Time-lapse imaging was performed under standard conditions (37 °C, 5% CO_2_), capturing four images per well using a ×20 objective lens, with images acquired every 2 h. Annexin V was imaged in the green fluorescence channel (300 ms exposure) and PI in the orange channel (400 ms exposure). Phase-contrast images were captured to segment the total cell population. Mia PaCa-2 cells were imaged for 36 h, while PL45 cells were imaged for 72 h. Quantification was performed using IncuCyte 2023A analysis software, and the percentage of Annexin V^+^/PI^+^ double-positive cells was calculated relative to the total segmented cell population at each time point.

### 4.8. Western Blotting

Western blot analysis was conducted for assessments of Caspase-8, -9, and -3 and PARP levels following treatment of PL45 cells (6 × 10^6^) with 175 µg/mL of *A. europaea* extract for 48 and 72 h, as well as Mia PaCa-2 cells (6 × 10^6^) with 150 µg/mL of the extract for 36 h. Control cells were treated with DMEM medium. At the end of treatment, cells were collected using trypsin and harvested by centrifugation at 2000 rpm for 5 min. Cells were washed once with 1 mL of cold PBSX1 and once again harvested by centrifugation at 2000 rpm for 5 min. For cell lyses, RIPA lyses buffer supplemented with protease inhibitor cocktail (Roche applied Science, Mannheim, Germany) was added to the cells. The mixture was kept on ice for 15 min, then cells were centrifuged at 2000 rpm for 15 min at 4 °C. Protein concentration was quantified by using the Bio-Rad Protein assay, based on Bradford’s method [38], using a BSA standard curve (0–25 µg/mL). Protein concentration was determined using a plate reader at 595 nm. Protein samples (60 µg) were diluted in sample buffer and heated at 95 °C for 5 min. Samples were separated on 10–15% SDS-polyacrylamide gels (SDS-PAGE) for 1.5 h at a power supply of 100–120 V, depending on the protein being tested. The protein samples were transferred from the gel to a PVDF membrane, using semi-dry transfer for 1 h at 15 V. The membrane was blocked in 20 mL of 5% non-fat dry milk in TBSTx1 buffer for 1 h at room temperature with shaking, then washed three times with TBSTx1. Next, the membrane was incubated overnight at 4 °C on a shaker in a blocking buffer containing primary antibodies against caspases-8 and -9 and PARP (Cell Signaling Technology, Danvers, MA, USA) and caspase-3 (Abcam, Cambridge, UK). Antibody dilutions were 1:1000 for caspases-8 and -9 and PARP and 1:5000 for caspase-3. Following three 10 min washes with TBSTX1, the membrane was incubated with a secondary antibody (Jakson Immuno Research Laboratories, West Grove, PA, USA) solution in TBSTX1 for 1 h at room temperature. Protein detection was performed using Chemiluminescent substrates—Luminol Enhancer and Peroxide solution reagents (Westar Antares, Cyanagen, Bologna, Italy), and images were acquired using ChemiDOc™XRS Gel documentation system (Amersham, Buckinghamshire, UK). Western blot results were quantified using Amersham Imager 600 analysis software v1.0.0. β-actin (Abcam, Cambridge, UK) was detected on the same membrane and used as a leading control.

### 4.9. Statistical Analysis

All experiments were repeated at least three times (unless indicated otherwise). All data were expressed as mean value ± standard error (SE), and the statistical differences between groups were evaluated using Student’s *t*-test for comparison between two groups or one-way analysis of variance (ANOVA) test for comparison between multiple groups. Two-way ANOVA followed by a post hoc test using Bonferroni adjustments was also employed to test the interactions of two sources of variation (time of treatment and concentration). *p* < 0.05 was considered statistically significant, and the SPSS software (IBM SPSS statistics 26) was used for the calculation of differences.

## Figures and Tables

**Figure 1 ijms-26-10221-f001:**
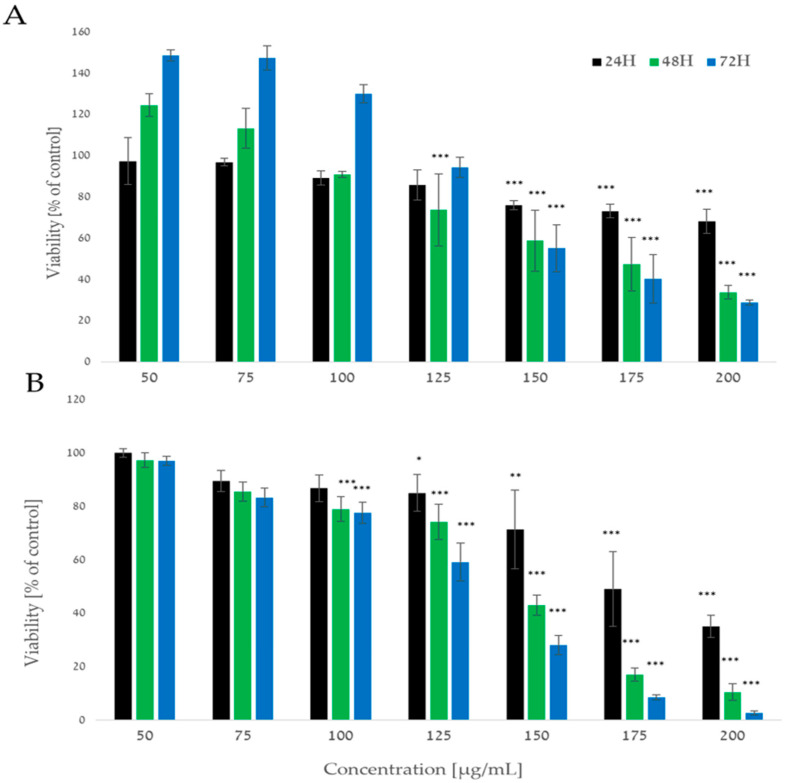
*A. europaea* extract reduces cell viability of human pancreatic cells. PL45 (**A**) and Mia PaCa-2 (**B**) cells were treated with increasing concentrations of *A. europaea* (50–200 µg/mL) for 24, 48, and 72 h and cell viability was determined using XTT assay. Data represent an average of three independent experiments; per experiment *n* = 3–5 repeats (mean ± SE) and are expressed as a percentage of the respective vehicle untreated control. Statistical significance was determined by one-way ANOVA test for post hoc to compare between concentrations and two-way ANOVA for time and concentration interactions. * *p* < 0.05, ** *p* < 0.01, and *** *p* < 0.001 indicate statistically significant differences compared with the untreated control.

**Figure 2 ijms-26-10221-f002:**
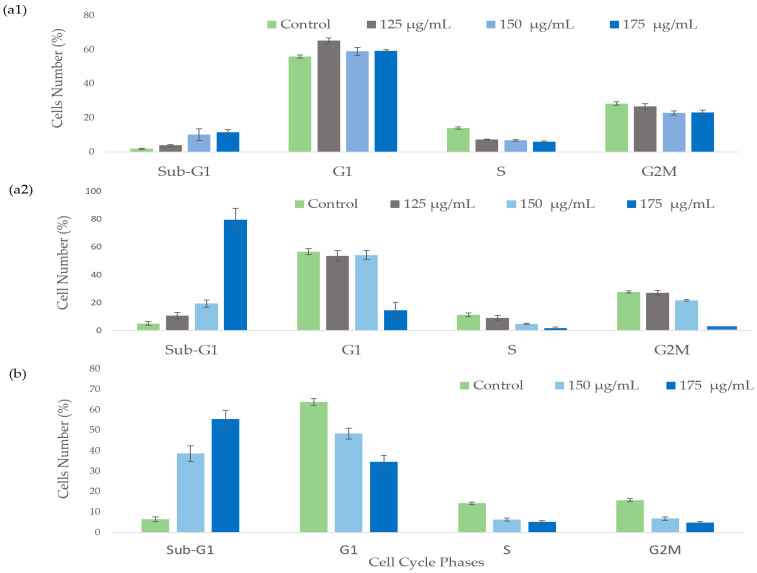
The effect of *A. europaea* extract on cell cycle distribution on PL-45 cell line (**a1**,**a2**) and on Mia PaCa-2 cell line (**b**). In total, 10^6^ PL-45 cells were treated with 125–175 μg/mL of *A. europaea* extract for 48 (**a1**) and 72 h (**a2**). A total of 10^6^ Mia PaCa-2 cells were treated with 150 and 175 μg/mL of *A. europaea* extract for 36 h (**b**). At the end of treatments, cells were harvested, fixed, and stained with PI. Quantitative analysis of DNA content in each phase was conducted by FACS. Data presented are average of three independent experiments (mean ± SE) and are expressed as percentages of the respective controls. Statistical significance was determined by two-way ANOVA for each cell line (*p* < 0.001).

**Figure 3 ijms-26-10221-f003:**
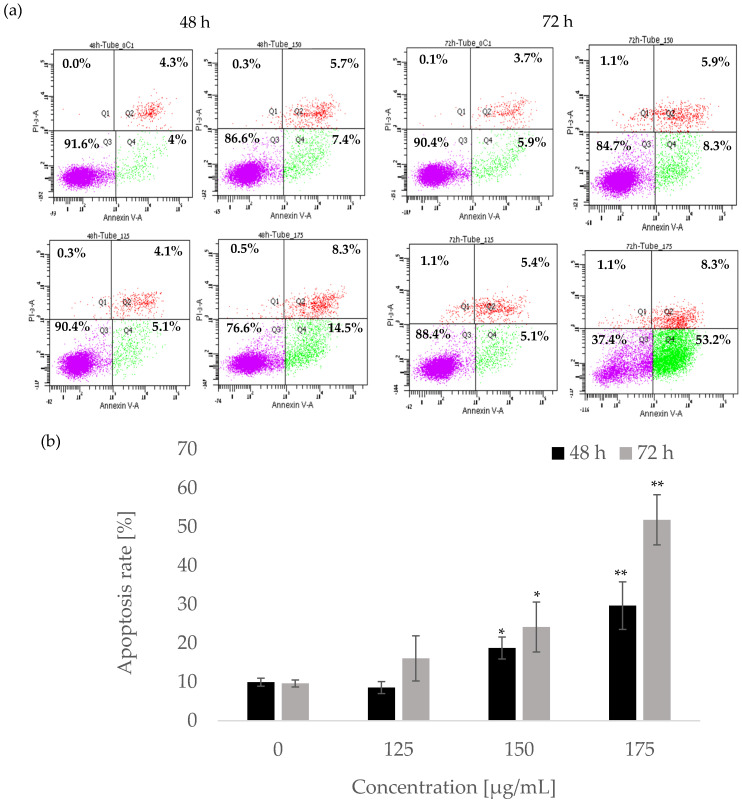
Measurement of apoptotic cells using Annexin V-FITC/PI double staining on PL45 (**a**,**b**) and Mia PaCa-2 (**c**,**d**) cells. In total, 10^6^ PL-45 cells were treated with either 125, 150, or 175 µg/mL of extract of *A. europaea* for 48 and 72 h. A total of 10^6^ Mia PaCa-2 cells were treated with either 150 or 175 µg/mL of the extract for 36 h and flow cytometric analysis of Annexin V-FITC/PI double staining was performed. In each plot (**a**,**c**), the lower left quadrant (Q3) represents viable cells, the upper left quadrant (Q1) indicates necrotic cells, the lower right quadrant (Q4) denotes early apoptotic cells, and the upper right quadrant (Q2) represents necrotic or late apoptosis cells. Data are presented as the mean ± SE of three independent experiments [mean (Q2 + Q4) ± SE] (**b**,**d**). Statistical significance was determined by a two-tailed Student’s *t*-test [treatment vs. control (untreated cells)] and is indicated as * *p* < 0.05, ** *p* < 0.01.

**Figure 4 ijms-26-10221-f004:**
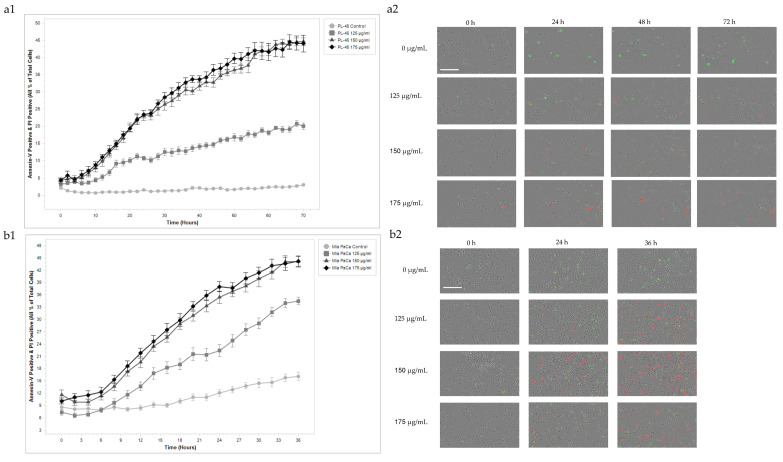
The IncuCyte live-cell analysis system used to study the effect of treatment with *A. europaea* extract on cell death in PL45 and Mia PaCa-2 cell lines. (**a1**) Real-time plots for cell death measurements in PL45 cells exposed to 125, 150, and 175 µg/mL of *A. europaea* extract for 72 h. (**a2**,**b2**) Representative images at X20 objective, of IncuCyte quantification in PL45 (**a2**) and Mia PaCa-2 (**b2**) cells. Scale bar 200 µm. In red, cells labeled with PI; in green, cells labeled with Annexin V. (**b1**) Real-time plots for cell death measurements in Mia PaCa-2 cells exposed to 125, 150, and 175 µg/mL of *A. europaea* extract for 36 h. Data are presented as mean ± SE of three independent experiments.

**Figure 5 ijms-26-10221-f005:**
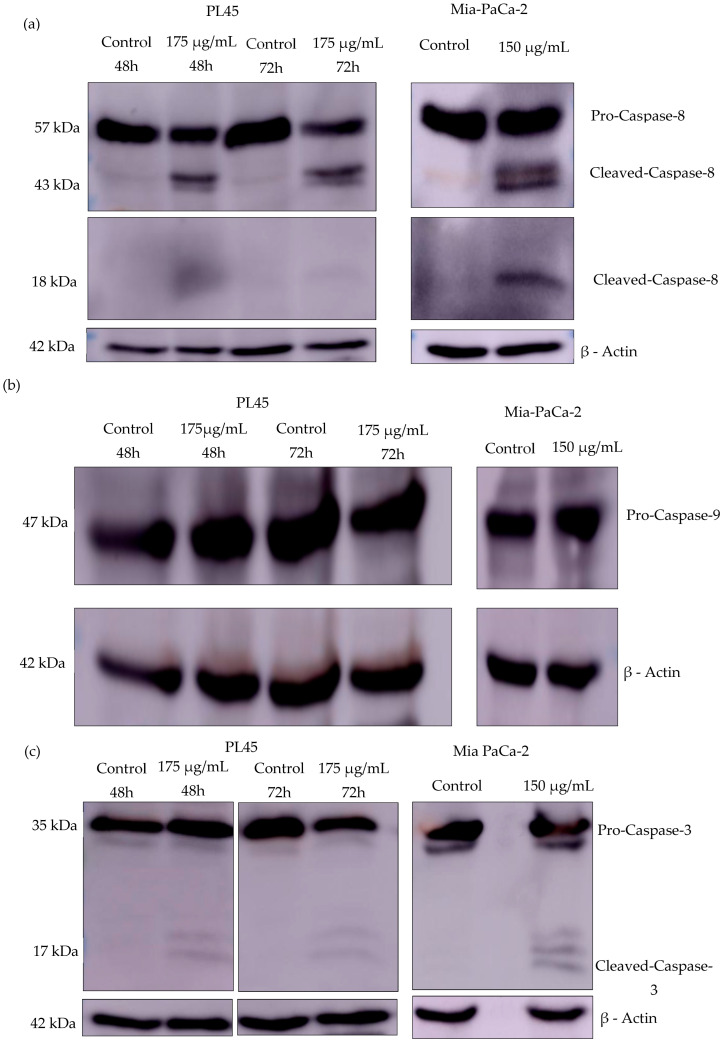
Western blot analysis of the caspases and PARP. PL45 and Mia PaCa-2 cells were treated with *A. europaea* extract. Mia PaCa-2 cells were treated with 150 µg/mL of *A. europaea* extract for 36 h, while PL45 cells were treated with 175 µg/mL for 48 and 72 h. The level of caspase-8 (**a**), caspase-9 (**b**), caspase-3 (**c**), and PARP (**d**) was analyzed by Western blotting, where β-actin was used as the loading control.

## Data Availability

The data presented in this study are available on request from the corresponding author.

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
