# Peer review of "Induction of Extrinsic Apoptotic Pathway in Pancreatic Cancer Cells by Apteranthes europaea Root Extract"

_ijms, 2025, doi:10.3390/ijms262010221_

Round 1
Reviewer 1 Report
Comments and Suggestions for Authors
After reading and evaluating the manuscript entitled "Antitumor Activity of Caralluma europaea Extract on Pancreatic Cancer Cells", I suggested some important modifications that should be considered by the authors to improve the quality of this study.
1. According to information available on the World Flora Online (WFO) Plant List platform (https://wfoplantlist.org/), the scientific name Caralluma europaea is a synonym of Apteranthes europaea (Guss.) Murb. Authors must include the correct and accepted scientific name of this species.
It is appropriate to replace “Caralluma europaea” with “Apteranthes europaea” throughout the text.
2. Lines 16-17: In the abstract methodology, authors must include the concentrations of C. europaea extract used to evaluate anticancer activity.
Additionally, authors should include the two pancreatic cancer cell lines used in this study (PL45 and MIA PaCa-2).
3. Line 22: Keywords should be replaced by others that do not appear in the manuscript title. This will increase the chances of this article being found in databases after publication.
4. Lines 27-31: Authors should mention which conventional drugs are currently available for the treatment of pancreatic cancer and their side effects.
This information is important to justify the development of new therapeutic alternatives based on medicinal plants with fewer side effects.
5. Figure 1: I suggest that the authors include the control data in Figure 1 to improve visualization and comparison between the extract results.
6. Figures 2 and 3: Why were PL-45 cells treated with C. europaea extract for 48 and 72 h and MIA PaCa-2 cells treated for only 36 h? Why was there no standardization in the duration of the experiment for both cell lines?
Why did the authors not standardize the extract concentrations for the tests in PL-45 cells (125, 150 and 175 μg/mL) and in MIA PaCa-2 cells (150 and 175 μg/mL)?
All of this must be detailed in the methodology of this study.
7. Lines 239-250: The first paragraph of the discussion includes a lot of general information from the scientific literature. I suggest deleting the first paragraph.
The discussion should be critical based on the results obtained in this study.
8. Lines 351-361: I identified some repeated information in the discussion text that should be checked and corrected by the authors.
The formatting of the manuscript text should also be improved.
9. Lines 367-369: Based on the results reported in this study, the authors should suggest the development of in vivo preclinical trials using experimental models of pancreatic cancer.
10. Line 373: Authors should include the month and year the species was collected.
11. Line 395: Why didn't the authors use a conventional drug used in cancer treatment as a positive control?
12. Lines 402-408: Was the cytotoxicity assay conducted in triplicate? Were negative and positive controls used? It is necessary to specify these details in the methodology.
13. Lines 410-411: In the Cell Cycle Analysis subsection, it is necessary to specify that the concentrations described refer to the plant extract.
14. Lines 410-411: The authors reported in the methodology that both cell lines were treated with the extract for 48 and 72 h. However, in Figure 2, cell cycle analysis of MIA PaCa-2 cells was analyzed for only 36 h. Authors should check this information.
15. It is necessary to conduct phytochemical analysis (HPLC or LC-MS) of the C. europaea root extract to identify the major chemical compounds responsible for the anticancer effects reported in this study.
Author Response
Reviewer 1
After reading and evaluating the manuscript entitled "Antitumor Activity of Caralluma europaea Extract on Pancreatic Cancer Cells", I suggested some important modifications that should be considered by the authors to improve the quality of this study.
- According to information available on the World Flora Online (WFO) Plant List platform (https://wfoplantlist.org/), the scientific name Caralluma europaea is a synonym of Apteranthes europaea (Guss.) Murb. Authors must include the correct and accepted scientific name of this species.
- It is appropriate to replace “Caralluma europaea” with “Apteranthes europaea” throughout the text.
We thank the reviewer for the comment, “Caralluma europaea” was replaced with “Apteranthes europaea” throughout the text.
- Lines 16-17: In the abstract methodology, authors must include the concentrations of C. europaea extract used to evaluate anticancer activity.
We thank the reviewer for the observation. The concentrations of C. europaea extract used in the anticancer activity assays have been added to the Abstract as requested.
- Additionally, authors should include the two pancreatic cancer cell lines used in this study (PL45 and MIA PaCa-2). –
We thank the reviewer for the suggestion. The two pancreatic cancer cell lines, PL45 and MIA PaCa-2, have been added to the Abstract to clarify the experimental model used in the study.
- Line 22: Keywords should be replaced by others that do not appear in the manuscript title. This will increase the chances of this article being found in databases after publication.
Thank you for the helpful comment. The keywords have been revised accordingly. The updated keywords are: phytochemicals, natural products, apoptosis induction, cytotoxicity, caspases, pancreatic cancer therapy.
- Lines 27-31: Authors should mention which conventional drugs are currently available for the treatment of pancreatic cancer and their side effects.
Thank you for the comments. conventional drugs available for the treatment of pancreatic cancer and their side effects was added to the text. “Conventional chemotherapy regimens currently used for PDAC include gemcitabine (often in combination with nab-paclitaxel) and FOLFIRINOX (a combination of 5-fluorouracil, leucovorin, irinotecan and oxaliplatin). For example, the combination of gemcitabine and nab-paclitaxel has shown improved response and overall survival compared to gemcitabine alone in locally advanced disease [2]. While these treatments can extend survival, they are associated with significant side effects including severe neutropenia, peripheral neuropathy, gastrointestinal toxicity (nausea, vomiting, diarrhea), fatigue, and in case of FOLFIRINOX, increased risk of febrile neutropenia and thrombocytopenia [3,4].”
This information is important to justify the development of new therapeutic alternatives based on medicinal plants with fewer side effects.
- Figure 1: I suggest that the authors include the control data in Figure 1 to improve visualization and comparison between the extract results.
Thank you for your valuable comment. The control was normalized to 100%, as it represents untreated cells. Therefore, including it in the figure would only result in a bar fixed at 100%, which we felt would not add additional information and might reduce the clarity of the visualization. Nevertheless, we fully acknowledge the importance of the control, and the raw data are surely available and can be provided if required.
- Figures 2 and 3: Why were PL-45 cells treated with C. europaea extract for 48 and 72 h and MIA PaCa-2 cells treated for only 36 h? Why was there no standardization in the duration of the experiment for both cell lines?
Thank you for this important comment. The Incucyte assay indicated that, for MIA PaCa-2 cells, the effect of C. europaea extract was already reached after 36 hours. In contrast, for PL-45 cells the response to treatment continued to evolve beyond this time point, which justified extending the assessment to 48 and 72 hours.
- Why did the authors not standardize the extract concentrations for the tests in PL-45 cells (125, 150 and 175 μg/mL) and in MIA PaCa-2 cells (150 and 175 μg/mL)?
We thank the reviewer for this important observation. The concentration range used for each cell line was based on preliminary dose–response experiments, which indicated different sensitivity levels between the two cell lines. PL45 cells showed a broader response range starting from 125 µg/mL, while MIA PaCa-2 cells exhibited minimal response at this lower concentration. Therefore, the experiments for MIA PaCa-2 focused on the more effective concentrations of 150 and 175 µg/mL to better capture the cytotoxic and apoptotic effects. This approach allowed us to tailor the treatment range to the biological response of each cell line and ensure more accurate analysis of the extract’s efficacy. All of this must be detailed in the methodology of this study..
- Lines 239-250: The first paragraph of the discussion includes a lot of general information from the scientific literature. I suggest deleting the first paragraph.
Thank you for the helpful comment. As suggested, the first paragraph of the Discussion section has been removed to improve focus and relevance.
- Lines 351-361: I identified some repeated information in the discussion text that should be checked and corrected by the authors.
We appreciate the reviewer’s observation. The Discussion section has been carefully reviewed, and the repeated information has been removed or rephrased to eliminate redundancy and improve clarity.
- The formatting of the manuscript text should also be improved.
We thank the reviewer for the suggestion. The manuscript text has been carefully revised to improve formatting for better readability and consistency throughout the document.
- Lines 367-369: Based on the results reported in this study, the authors should suggest the development of in vivo preclinical trials using experimental models of pancreatic cancer.
We thank the reviewer for this valuable suggestion. Accordingly, we have added the following statement to the manuscript:
“Based on these promising in vitro results, future research should focus on developing in vivo preclinical trials using experimental models of pancreatic cancer to further validate the therapeutic potential of this extract.”
- Line 373: Authors should include the month and year the species was collected.
We thank the reviewer for the suggestion. The date of plant collection has been added to the manuscript: “The species was collected in June 2023.”
- Line 395: Why didn't the authors use a conventional drug used in cancer treatment as a positive control?
We appreciate the reviewer's important observation regarding the use of a conventional anticancer drug as a positive control. We acknowledge that including a standard chemotherapeutic agent (such as gemcitabine, which is a first-line treatment for pancreatic cancer, or 5-fluorouracil) would have strengthened our experimental design and provided a valuable reference point for comparing the efficacy of C. europaea root extract.
In this initial screening study, our primary objective was to establish the baseline cytotoxic activity of the plant extract and elucidate its mechanism of action. However, we recognize that the absence of a conventional positive control limits our ability to contextualize the therapeutic potential of our extract relative to established treatments.
We will incorporate appropriate positive controls (such as gemcitabine or other clinically relevant anticancer agents) in future studies to provide a more comprehensive evaluation of the extract's anticancer efficacy. This will enable direct comparison with standard treatments and better assessment of the extract's potential as an alternative or adjuvant therapeutic approach.
This limitation has been noted and will be addressed in our ongoing research to strengthen the translational relevance of our findings.
- Lines 402-408: Was the cytotoxicity assay conducted in triplicate? Were negative and positive controls used? It is necessary to specify these details in the methodology.
We thank the reviewer for this important methodological clarification. Yes, the cytotoxicity assay (LDH) was conducted in triplicate, as specified in Appendix A1 in the figure legend, which states: "The data presented are an average of three independent experiments each conducted in triplicates (mean ± SE) and are expressed as percentages of respective controls."
Regarding controls, negative controls (untreated cells) were included in all experiments, as indicated by the control groups shown in our figures. However, we acknowledge that we did not include positive controls using conventional anticancer drugs in this initial study, which represents a limitation of our experimental design.
To address this concern and improve clarity, we will revise the methodology section to explicitly state the experimental design, including the use of triplicates and negative controls: “Untreated cells served as negative controls. 24 h post treatment, the levels of LDH in the cell culture media were detected by the CyQUANTTM LDH Cytotoxicity Assay (Invitrogen, North America) following the manufacturer’s instructions. All experiments were conducted in triplicate, and data are presented as the average of three independent experiments each conducted in triplicates (mean ± SE) and expressed as percentages of respective controls”.
For future studies, we will incorporate appropriate positive controls (such as gemcitabine or other clinically relevant anticancer agents) to provide better context for evaluating the therapeutic potential of the C. europaea extract relative to established treatments.
- Lines 410-411: In the Cell Cycle Analysis subsection, it is necessary to specify that the concentrations described refer to the plant extract.
The concentrations of the plant extract on cell cycle was added to the text. “Cells were treated with A. europaea root extract at concentrations of 125, 150 and 175 µg/mL for PL45 and with 150 and 175 µg/mL for MIA PaCa-2 cells, for 48 and 72 h for PL45 and 36 h for MIA PaCa-2 cells.
- Lines 410-411: The authors reported in the methodology that both cell lines were treated with the extract for 48 and 72 h. However, in Figure 2, cell cycle analysis of MIA PaCa-2 cells was analyzed for only 36 h. Authors should check this information.
The revised text specifies in the methodology section that PL45 cells were treated for 48 and 72 hours, while MIA PaCa-2 cells were treated for 36 hours.
- It is necessary to conduct phytochemical analysis (HPLC or LC-MS) of the C. europaea root extract to identify the major chemical compounds responsible for the anticancer effects reported in this study.
The phytochemistry studies was already carried out on the aerial parts of the plant, while the root characterization is still underway. This will take some time for us, as our focus is on identifying the novel metabolites rather than repeating what has already been reported for the plant. Please see the paragraph in the discussion: From a phytochemical point of view, Apteranthes europaea has been well-documented. Studies on different parts of the plant, using solvents like methanol, hydroalcoholic mixtures, and ethyl acetate, have consistently shown it to possess a rich and diverse profile of compounds, primarily phenolics. Chromatographic analyses of various extracts have identified a wide array of flavonoids, including luteolin and its glycosylated derivatives (luteolin-3′,4′-O-diglucoside, luteolin-4′-O-neohesperidoside, luteolin-7-O-glucoside), quercetin (both aglycone and as quercetin-3-O-rutinoside/rutin), kaempferol (and its derivative kaempferol-3-O-hexose deoxyhexose), myricetin, apigenin-4′-O-neohesperidoside, hesperetin, catechin, and epicatechin. The phenolic acid composition is equally broad, encompassing gallic acid, trans-ferulic acid, caffeic acid, chlorogenic acid, syringic acid, salicylic acid, p-coumaric acid, rosmarinic acid, vanillic acid, 3,4-dihydroxybenzoic acid, 2-hydroxycinnamic acid, sinapic acid, and ascorbic acid. This consistent identification of bioactive phenolics across hydroethanolic, ethyl acetate, and methanolic extracts firmly establishes C. europaea as a significant source of these compounds, providing a chemical basis for its investigated biological activities [6,19,21,23,24]

Reviewer 2 Report
Comments and Suggestions for Authors
Review
General Overview
The manuscript presents a study on the anticancer effects of Caralluma europaea root extract against pancreatic ductal adenocarcinoma (PDAC) cell lines (PL45 and MIA PaCa-2). The goal of this study is relevant considering the need for novel therapeutic strategies for PDAC, bearing in mind limited treatment options and poor prognosis. The study elucidates both the cytotoxic effects and mechanistic pathways of apoptosis activation by C. europaea root extract treatment. The applied methodology, e.g., cell viability assays, cell cycle analysis, apoptosis assays (Annexin V/PI, live-cell imaging), and Western blotting, contribute to the validation of presented results.
Presentation and Language
Overall, the manuscript is well-written. A few changes in terminology (e.g., "medicinal medicine" in text line 36) and lapsus calami, as in line 47 (anti-inflammatory ,bone), lines 189–106, copmositions (line 259), and “sug-G1” in text line 300, should be corrected.
Major Concerns
- Phytochemical composition of the root extract:
The manuscript lacks a chemical characterisation of the used extract. Identification of major constituents would significantly strengthen the conclusions and reproducibility.
- Selectivity of anticancer effects:
The study does not provide information on the effects of the extract on normal pancreatic or non-cancerous epithelial cells, which is essential for the evaluation of its therapeutic potential.
- Lines in the manuscript from 351 to 361 do not belong to the manuscript.
Limitations
The manuscript could be made more concise and engaging for journal impact.
The present work does not provide a phytochemical profile of the root extract. Without such analysis, it is difficult to link the reported effects to actual chemicals that are present in the extract.
Conclusion and Recommendation
The study contributes to the field of natural product-based cancer therapeutics by demonstrating that C. europaea root extract inhibits PDAC cell growth and induces apoptosis through the extrinsic pathway. However, for potential publication manuscript should include LC-MS/MS or HPLC profiling to identify and quantify major bioactive constituents responsible for the pro-apoptotic activity. Although LDH assay suggest lack of nonspecific necrosis, the study does not test the extract’s effects on non-cancerous pancreatic or normal epithelial cells, which is necessary for assessing selectivity and potential therapeutic window. Please provide cytotoxicity (XTT, LDH) testing on non-cancerous pancreatic or normal epithelial cells or reasonable explanation for lacking data.
Recommendation:
Accept with major revisions focusing on:
- cytotoxicity (XTT, LDH) testing on non-cancerous pancreatic or normal epithelial cells or provide reasonable explanation for lacking data.
- chemical profiling of extract.
- language polishing.
As stated in the Manuscript Preparation under General Considerations
Important note:
Substances without clear ingredients, such as complex prescriptions, crude extracts, and herbal mixtures, are not considered.
Reviewers’ comments
Title of the manuscript should be modified. "Antitumor activity" is broad; the paper specifically shows apoptosis via the extrinsic pathway, which is more precise.
The introduction is well written, but the sentence (lines 55 and 56) could be rewritten for clarity and readability.
Results
The sentence "In order to exclude the possibility of cytotoxic effects of C. europaea extract on the cells, lactate dehydrogenase (LDH) leakage assay was performed", should be rewritten since the authors showed by the XTT assay that treatments with higher concentrations and prolonged treatment time decreased cell viability, pointing to the cytotoxic impact on both cell lines. "Possibility of cytotoxic effects" could be modified to "excessive necrotic cell death", or similar.
Figures should have uniform x- and y-axis title font sizes. Figure 1 has a capital H mark for hour instead of h. Figures at the end of the manuscript slightly differ from the ones in text Figure 2 (axis titles font size and position) and Figure 5 (marked as Figure 4 and not represented as in text). As stated in the instruction for authors, images of cells and western blots should be large enough to see the relevant features. In addition, uncropped, untouched, full original images of western blots should be uploaded with the other figure files. Please provide previously mentioned figures.
Explain why were MIA PaCa-2 cells treated with C. europaea extract for 36 h? Previous analyses were done at different time points (24, 48 and 72 h).
Discussion
The sentence "The use of different parts of the plant can lead to distinct chemical copmositions, such as polyphenols and flavonoids, which exhibit anticancer properties, and consequently, different outcomes. " is not clear and should be rewritten. References 24-26 could be quoted but not detailed.
In the discussion section, authors should refer to literature data that have a strong connection to their research. The following part of the discussion section is not strongly connected and should be excluded from the manuscript:
"These defects contribute to poor treatment response and tumor survival. Therefore, the ability of C. europaea extract to trigger the death receptor-mediated pathway represents a significant therapeutic advantage. Activation of caspase-8 implies involvement of upstream death receptors, such as TRAIL-R1/R2 or Fas, which may remain functionally accessible in these cells despite other resistance mechanisms [37], although the specific receptors and upstream ligands involved remain to be identified. Another study reported the chemical composition of the aerial parts of C. europea using time-of-flight mass spectrometry and high-performance liquid chromatography. The analysis revealed the presence of several lipids, including linoleic acid and vitamin D₃. Among the identified compounds, lignoceric acid demonstrated the highest activity against caspase-3, a cysteine protease that plays a crucial role in apoptosis [38]. "
Conclusion
The sentence "These findings not only expand the known pharmacological potential of C. europaea but also highlight the importance of investigating underexplored plant species in the search for novel anticancer agents" should be removed since the pharmacological potential of C. europaea was investigated in the first place due to the common knowledge that secondary metabolites of the plant species could be important in the search for novel anticancer agents.
Methods
A range of complementary assays (XTT viability, LDH release, flow cytometry, IncuCyte live-cell imaging, Annexin V/PI staining, and Western blotting) and statistical analyses (two-way ANOVA, post-hoc testing, R² values), contributes to the validation and the reliability of the data.
References
References should be presented as in the instructions for the authors.
Comments on the Quality of English LanguageOverall, the manuscript is well-written. A few changes in terminology (e.g., "medicinal medicine" in text line 36) and lapsus calami, as in line 47 (anti-inflammatory ,bone), lines 189–106, copmositions (line 259), and “sug-G1” in text line 300, should be corrected.
Some sentences could be rewritten for clarity, as stated in the reviewers’ comments.
Author Response
Reviewer 2
General Overview
The manuscript presents a study on the anticancer effects of Caralluma europaea root extract against pancreatic ductal adenocarcinoma (PDAC) cell lines (PL45 and MIA PaCa-2). The goal of this study is relevant considering the need for novel therapeutic strategies for PDAC, bearing in mind limited treatment options and poor prognosis. The study elucidates both the cytotoxic effects and mechanistic pathways of apoptosis activation by C. europaea root extract treatment. The applied methodology, e.g., cell viability assays, cell cycle analysis, apoptosis assays (Annexin V/PI, live-cell imaging), and Western blotting, contribute to the validation of presented results.
Presentation and Language
Overall, the manuscript is well-written. A few changes in terminology (e.g., "medicinal medicine" in text line 36) and lapsus calami, as in line 47 (anti-inflammatory ,bone), lines 189–106, copmositions (line 259), and “sug-G1” in text line 300, should be corrected.
Major Concerns
- Phytochemical composition of the root extract:
The manuscript lacks a chemical characterisation of the used extract. Identification of major constituents would significantly strengthen the conclusions and reproducibility.
The phytochemistry studies was already carried out on the aerial parts of the plant, while the root characterization is still underway. This will take some time for us, as our focus is on identifying the novel metabolites rather than repeating what has already been reported for the plant. Please see the paragraph in the discussion: “From a phytochemical point of view, Apteranthes europaea has been well-documented. Studies on different parts of the plant, using solvents like methanol, hydroalcoholic mixtures, and ethyl acetate, have consistently shown it to possess a rich and diverse profile of compounds, primarily phenolics. Chromatographic analyses of various extracts have identified a wide array of flavonoids, including luteolin and its glycosylated derivatives (luteolin-3′,4′-O-diglucoside, luteolin-4′-O-neohesperidoside, luteolin-7-O-glucoside), quercetin (both aglycone and as quercetin-3-O-rutinoside/rutin), kaempferol (and its derivative kaempferol-3-O-hexose deoxyhexose), myricetin, apigenin-4′-O-neohesperidoside, hesperetin, catechin, and epicatechin. The phenolic acid composition is equally broad, encompassing gallic acid, trans-ferulic acid, caffeic acid, chlorogenic acid, syringic acid, salicylic acid, p-coumaric acid, rosmarinic acid, vanillic acid, 3,4-dihydroxybenzoic acid, 2-hydroxycinnamic acid, sinapic acid, and ascorbic acid. This consistent identification of bioactive phenolics across hydroethanolic, ethyl acetate, and methanolic extracts firmly establishes C. europaea as a significant source of these compounds, providing a chemical basis for its investigated biological activities [6,19,21,23,24]”
- Selectivity of anticancer effects:
The study does not provide information on the effects of the extract on normal pancreatic or non-cancerous epithelial cells, which is essential for the evaluation of its therapeutic potential.
This study did not assess the effects of the plant extract on normal (non-cancerous) cells. However, to preliminarily evaluate its cytotoxicity, an LDH assay was performed, indicating that the extract does not exhibit toxic effects under the tested conditions. Future studies will include in vivo toxicity assessments in animal models to further validate its safety profile.
- Lines in the manuscript from 351 to 361 do not belong to the manuscript.
We thank the reviewer for pointing this out. The paragraph in lines 351 to 361 has been removed and the section has been revised accordingly to maintain relevance and coherence.Limitations
The manuscript could be made more concise and engaging for journal impact.
- The present work does not provide a phytochemical profile of the root extract. Without such analysis, it is difficult to link the reported effects to actual chemicals that are present in the extract.
The phytochemistry studies was already carried out on the aerial parts of the plant, while the root characterization is still underway. This will take some time for us, as our focus is on identifying the novel metabolites rather than repeating what has already been reported for the plant. Please see the paragraph in the discussion: “From a phytochemical point of view, Apteranthes europaea has been well-documented. Studies on different parts of the plant, using solvents like methanol, hydroalcoholic mixtures, and ethyl acetate, have consistently shown it to possess a rich and diverse profile of compounds, primarily phenolics. Chromatographic analyses of various extracts have identified a wide array of flavonoids, including luteolin and its glycosylated derivatives (luteolin-3′,4′-O-diglucoside, luteolin-4′-O-neohesperidoside, luteolin-7-O-glucoside), quercetin (both aglycone and as quercetin-3-O-rutinoside/rutin), kaempferol (and its derivative kaempferol-3-O-hexose deoxyhexose), myricetin, apigenin-4′-O-neohesperidoside, hesperetin, catechin, and epicatechin. The phenolic acid composition is equally broad, encompassing gallic acid, trans-ferulic acid, caffeic acid, chlorogenic acid, syringic acid, salicylic acid, p-coumaric acid, rosmarinic acid, vanillic acid, 3,4-dihydroxybenzoic acid, 2-hydroxycinnamic acid, sinapic acid, and ascorbic acid. This consistent identification of bioactive phenolics across hydroethanolic, ethyl acetate, and methanolic extracts firmly establishes C. europaea as a significant source of these compounds, providing a chemical basis for its investigated biological activities [6,19,21,23,24]”
Conclusion and Recommendation
The study contributes to the field of natural product-based cancer therapeutics by demonstrating that C. europaea root extract inhibits PDAC cell growth and induces apoptosis through the extrinsic pathway. However, for potential publication manuscript should include LC-MS/MS or HPLC profiling to identify and quantify major bioactive constituents responsible for the pro-apoptotic activity. Although LDH assay suggest lack of nonspecific necrosis, the study does not test the extract’s effects on non-cancerous pancreatic or normal epithelial cells, which is necessary for assessing selectivity and potential therapeutic window. Please provide cytotoxicity (XTT, LDH) testing on non-cancerous pancreatic or normal epithelial cells or reasonable explanation for lacking data.
This study did not assess the effects of the plant extract on normal (non-cancerous) cells. However, to preliminarily evaluate its cytotoxicity, an LDH assay was performed, indicating that the extract does not exhibit toxic effects under the tested conditions. Future studies will include in vivo toxicity assessments in animal models to further validate its safety profile.
Recommendation:
Accept with major revisions focusing on:
- cytotoxicity (XTT, LDH) testing on non-cancerous pancreatic or normal epithelial cells or provide reasonable explanation for lacking data.
This study did not assess the effects of the plant extract on normal (non-cancerous) cells. However, to preliminarily evaluate its cytotoxicity, an LDH assay was performed, indicating that the extract does not exhibit toxic effects under the tested conditions. Future studies will include in vivo toxicity assessments in animal models to further validate its safety profile.
- chemical profiling of extract.
The phytochemistry studies was already carried out on the aerial parts of the plant, while the root characterization is still underway. This will take some time for us, as our focus is on identifying the novel metabolites rather than repeating what has already been reported for the plant. Please see the paragraph in the discussion: “From a phytochemical point of view, Apteranthes europaea has been well-documented. Studies on different parts of the plant, using solvents like methanol, hydroalcoholic mixtures, and ethyl acetate, have consistently shown it to possess a rich and diverse profile of compounds, primarily phenolics. Chromatographic analyses of various extracts have identified a wide array of flavonoids, including luteolin and its glycosylated derivatives (luteolin-3′,4′-O-diglucoside, luteolin-4′-O-neohesperidoside, luteolin-7-O-glucoside), quercetin (both aglycone and as quercetin-3-O-rutinoside/rutin), kaempferol (and its derivative kaempferol-3-O-hexose deoxyhexose), myricetin, apigenin-4′-O-neohesperidoside, hesperetin, catechin, and epicatechin. The phenolic acid composition is equally broad, encompassing gallic acid, trans-ferulic acid, caffeic acid, chlorogenic acid, syringic acid, salicylic acid, p-coumaric acid, rosmarinic acid, vanillic acid, 3,4-dihydroxybenzoic acid, 2-hydroxycinnamic acid, sinapic acid, and ascorbic acid. This consistent identification of bioactive phenolics across hydroethanolic, ethyl acetate, and methanolic extracts firmly establishes C. europaea as a significant source of these compounds, providing a chemical basis for its investigated biological activities [6,19,21,23,24]”
- language polishing.
As stated in the Manuscript Preparation under General Considerations
Important note:
Substances without clear ingredients, such as complex prescriptions, crude extracts, and herbal mixtures, are not considered.
The manuscript was prepared to a special issue of “natural products and cancer”. On the other hands the the phytochemistry studies was already carried out on the aerial parts of the plant. Chromatographic analyses of various extracts have identified a wide array of flavonoids, including luteolin and its glycosylated derivatives (luteolin-3′,4′-O-diglucoside, luteolin-4′-O-neohesperidoside, luteolin-7-O-glucoside), quercetin (both aglycone and as quercetin-3-O-rutinoside/rutin), kaempferol (and its derivative kaempferol-3-O-hexose deoxyhexose), myricetin, apigenin-4′-O-neohesperidoside, hesperetin, catechin, and epicatechin. The phenolic acid composition is equally broad, encompassing gallic acid, trans-ferulic acid, caffeic acid, chlorogenic acid, syringic acid, salicylic acid, p-coumaric acid, rosmarinic acid, vanillic acid, 3,4-dihydroxybenzoic acid, 2-hydroxycinnamic acid, sinapic acid, and ascorbic acid. This consistent identification of bioactive phenolics across hydroethanolic, ethyl acetate, and methanolic extracts firmly establishes C. europaea as a significant source of these compounds, providing a chemical basis for its investigated biological activities [6,19,21,23,24]
Reviewers’ comments
Title of the manuscript should be modified. "Antitumor activity" is broad; the paper specifically shows apoptosis via the extrinsic pathway, which is more precise.
We thank the reviewer for this helpful suggestion. To better reflect the specific findings of the study, the manuscript title has been revised to:
“Induction of the Extrinsic Apoptotic Pathway in Pancreatic Cancer Cells by Apteranthes europaea Root Extract.”
The introduction is well written, but the sentence (lines 55 and 56) could be rewritten for clarity and readability.
to the sentence in the introduction was revised: Additionally, this study demonstrated that the hydroethanolic extract from A. europaea aerial parts of, enriched with flavonoids, polyphenols, and saponin effectively reduced the expression of pancreatic cancer associated markers – Lines 65-67
Results
The sentence "In order to exclude the possibility of cytotoxic effects of C. europaea extract on the cells, lactate dehydrogenase (LDH) leakage assay was performed", should be rewritten since the authors showed by the XTT assay that treatments with higher concentrations and prolonged treatment time decreased cell viability, pointing to the cytotoxic impact on both cell lines. "Possibility of cytotoxic effects" could be modified to "excessive necrotic cell death", or similar.
Lines 90-91 were changed to: To assess whether A. europaea extract treatments induced excessive membrane damage and necrtotic cell death, Lactate dehydrogenase (LDH) leakage assay was performed.
And lines 614-615 in the methods section were changed to: To distinguish between cytotoxic effect and excessive necrotic cell death following A. europaea extract treatments, Lactate dehydrogenase (LDH) leakage assay was performed
Figures should have uniform x- and y-axis title font sizes. Figure 1 has a capital H mark for hour instead of h. Figures at the end of the manuscript slightly differ from the ones in text Figure 2 (axis titles font size and position) and Figure 5 (marked as Figure 4 and not represented as in text). As stated in the instruction for authors, images of cells and western blots should be large enough to see the relevant features. In addition, uncropped, untouched, full original images of western blots should be uploaded with the other figure files. Please provide previously mentioned figures.-
The Figures were changed according to the reviewer comments
Explain why were MIA PaCa-2 cells treated with C. europaea extract for 36 h? Previous analyses were done at different time points (24, 48 and 72 h).
Thank you for this important comment. The Incucyte assay indicated that, for MIA PaCa-2 cells, the effect of C. europaea extract was already reached after 36 hours. In contrast, for PL-45 cells the response to treatment continued to evolve beyond this time point, which justified extending the assessment to 48 and 72 hours.
Discussion
The sentence "The use of different parts of the plant can lead to distinct chemical copmositions, such as polyphenols and flavonoids, which exhibit anticancer properties, and consequently, different outcomes. " is not clear and should be rewritten. References 24-26 could be quoted but not detailed.
Thank you for this feedback. We have rewritten the sentence for clarity and condensed the discussion of references 24-26 as requested. The revised paragraph now more clearly explains how different plant parts lead to variable anticancer activities.
Here is the revised paragraph: Lines 467-479
Different plant parts contain varying concentrations of bioactive compounds, resulting in distinct anticancer activities. Plant [arts such as leaves, stems, flowers and roots can differ significantly im their polyphenol and flavonoid content, leading to variable therapeutic outcomes. This variation in efficacy has been demonstrated across multiple studies [25–27]. For instance, hydromethanolic extract from Bauhinia variegata floral buds showed superior antitumor activity against melanoma compared to leaf and stem bark extract [25]. Similarly, aqueous leaf extract of Inula viscosa demonstrated significant anticancer effect against colorectal cancer cells in both in vitro and in vivo models [26]. However, this varizbility can also result in inconsistent outcomes, as demonstrated by Anglana et al. [27], who found that aqueous extracts from aerial parts of Inula viscosa exhibited moderate cytotoxicity against SW620 colorectal cancer cells but had no significant effect on DLD-1 and HT-29 cell lines.
In the discussion section, authors should refer to literature data that have a strong connection to their research. The following part of the discussion section is not strongly connected and should be excluded from the manuscript:
"These defects contribute to poor treatment response and tumor survival. Therefore, the ability of C. europaea extract to trigger the death receptor-mediated pathway represents a significant therapeutic advantage. Activation of caspase-8 implies involvement of upstream death receptors, such as TRAIL-R1/R2 or Fas, which may remain functionally accessible in these cells despite other resistance mechanisms [37], although the specific receptors and upstream ligands involved remain to be identified. Another study reported the chemical composition of the aerial parts of C. europea using time-of-flight mass spectrometry and high-performance liquid chromatography. The analysis revealed the presence of several lipids, including linoleic acid and vitamin D₃. Among the identified compounds, lignoceric acid demonstrated the highest activity against caspase-3, a cysteine protease that plays a crucial role in apoptosis [38]. "
Thank you for the comment, this paragraph was excluded from the discussion.
Conclusion
The sentence "These findings not only expand the known pharmacological potential of C. europaea but also highlight the importance of investigating underexplored plant species in the search for novel anticancer agents" should be removed since the pharmacological potential of C. europaea was investigated in the first place due to the common knowledge that secondary metabolites of the plant species could be important in the search for novel anticancer agents.
Thank you for your comment. In accordance with your suggestion, the sentence has been excluded from the conclusionMethods
A range of complementary assays (XTT viability, LDH release, flow cytometry, IncuCyte live-cell imaging, Annexin V/PI staining, and Western blotting) and statistical analyses (two-way ANOVA, post-hoc testing, R² values), contributes to the validation and the reliability of the data.
References
References should be presented as in the instructions for the authors.
The references were presented according to the journal instruction
Comments on the Quality of English Language
Overall, the manuscript is well-written. A few changes in terminology (e.g., "medicinal medicine" in text line 36) and lapsus calami, as in line 47 (anti-inflammatory ,bone), lines 189–106, copmositions (line 259), and “sug-G1” in text line 300, should be corrected.
- "medicinal medicine" → "medicinal purposes" – Line 46
- "praticularly" → "particularly" – Line 47
- "medical plant" → "medicinal plant"- Line 53
- "anti-inflammatory ,bone" → "anti-inflammatory, bone disorders,"- Line 57
- Added missing comma after "antifungal "- Line 59
Some sentences could be rewritten for clarity, as stated in the reviewers’ comments.
Corrections were made according to the reviewer comments:

Reviewer 3 Report
Comments and Suggestions for Authors
The manuscript describes the antitumor activity of Caralluma europaea root extract on pancreatic cells. The topic is interesting because roots of the plant are used and not the areal part. However, some points should be clarified and improved.
-The composition of extract from the areal part has been analysed and published. What about the root extract? Is it known?
-Methanol was used for the extraction and then the extract was dissolved in absolute ethanol prior usage. The effects of the methanol/alcohol should be taken into account. Thus, authors should perform the experiments by using the corresponding vehicle as control and not cells in culture medium.
-Authors should evaluate the IC50 at each time point in the viability assay and they should test the root extract also in normal cells.
- After the viability test, the authors used the doses of 150 and 175 µg/mL for the subsequent experiments. Doses selected after 48 h of treatment for MIA-PaCa-2 cells. Why then the authors change the treatment time from 48 hours to 36 hours in the experiments for MIA-PaCa-2 cells?
- The FACS graphs are all completely blurry. You can't read a word. It is necessary to improve the quality.
-Authors declared that no cleavage of pro-caspase 9 is evident. However, it is not possible to make this conclusion since the western blot Figure 5 panel B is cut to show only the procaspase form and not any cleaved fragments which have a lower molecular weight (37/35 and 12/10 kDa). Authors should clarify.
- In Figure 5, panel 5, MIA-Paca-2 cells, the Western blot image shows an irregular distance between the two control and treatment lanes. What happened to this gel?
-To better strengthen the conclusion, the analysis of the mitochondria membrane potential should be shown.
-Authors should revise the section from line 351 to 360. It is not the right place.
- It would be interesting to see if the extract is able to function as an anti-tumor agent even in pancreatic cancer cells resistant to conventional chemotherapy.
Author Response
Reviewer 3
Introduction
- The Introduction needs more improvement. The main idea of the manuscript is not clearly explained, and the points are not well connected. The authors should give more background on pancreatic cancer, explain why natural products are important in cancer research, and clearly state why Caralluma europaeawas chosen. This will help the reader better understand the purpose and importance of the study.
We thank the reviewer for this valuable comment. In response, the Introduction has been substantially revised to improve clarity and logical flow. Additional background on pancreatic cancer has been included, along with a discussion of the significance of natural products in cancer therapy. Furthermore, the rationale for selecting Caralluma europaea for this study has been clearly stated. These changes aim to better frame the research question and highlight the relevance and importance of the study.
Result
- In section 2.1, only PL45 and MIA-PaCa-2 pancreatic cancer cells were used. The study would be stronger if the authors also included a normal cell line as a control.
This study did not assess the effects of the plant extract on normal (non-cancerous) cells. However, to preliminarily evaluate its cytotoxicity, an LDH assay was performed, indicating that the extract does not exhibit toxic effects under the tested conditions. Future studies will include in vivo toxicity assessments in animal models to further validate its safety profile.
- All figures should be presented in higher resolution, with clearer labeling and improved color contrast.
We thank the reviewer for this helpful comment. All figures have been updated to higher-resolution versions with improved labeling and enhanced color contrast to ensure better clarity and readability, as recommended.
- Lines 351–361 are not well organized and need to be rewritten for clarity
As suggested by the reviewer, the paragraph has been clarified to enhance clarity and improve understanding.
- The study currently provides only in vitro evidence. There is no information about control cells, which makes the results less convincing.
We appreciate the reviewer’s comment. This study focuses on investigating the effect of the plant extract on the growth of pancreatic cancer cells in vitro. We acknowledge the limitation of not including control (normal) cells in this initial study. Future work will include in vivo experiments using animal models to evaluate the extract’s effects on tumor growth, as well as its safety profile in normal tissues.
- More experimental data are needed, for example qRT-PCR analysis to support the in vitro findings.
We appreciate the reviewer’s suggestion. In future studies, we plan to perform qRT-PCR analysis to further investigate and confirm the molecular mechanisms underlying the plant extract’s role in inducing apoptosis. These additional experiments will help to strengthen and validate the current in vitro findings.
- An in vivo study should also be included to strengthen the manuscript and validate the anticancer potential of Caralluma europaea.
We thank the reviewer for this valuable suggestion. Future studies are planned to evaluate the anticancer efficacy of A. europaea extract in vivo using appropriate animal models. These studies will help to validate the current in vitro findings and provide deeper insight into the extract’s therapeutic potential.
Discussion
- The pathway should be prepared more clearly
We appreciate the reviewer’s comment. The molecular pathway has been clarified and more clearly presented in the Discussion section to improve understanding of the proposed mechanism.
- Materials and Methods
- Section 4.1: The heading mentions “Preparations of Mushroom Extracts,” but the study is about Caralluma europaea. This needs clarification and correction. The authors should also provide details about the chemical composition of the C. europaea extract, supported by techniques such as HPLC and GC-MS analysis. More plant-based experiments are required to strengthen the data.
We thank the reviewer for this important observation. The heading was mistakenly labeled and has now been corrected to accurately reflect the content of the study, which investigates the efficacy of Caralluma europaea extract on pancreatic cancer cells. Previous chromatographic analyses of A. europaea extracts—conducted using HPLC and GC-MS techniques—have identified a wide spectrum of flavonoids and phenolic acids. Notably, these include luteolin and its glycosylated derivatives (e.g., luteolin-3′,4′-O-diglucoside, luteolin-4′-O-neohesperidoside, luteolin-7-O-glucoside), quercetin (including quercetin-3-O-rutinoside/rutin), kaempferol (and its glycosylated forms), myricetin, apigenin-4′-O-neohesperidoside, hesperetin, catechin, and epicatechin. The phenolic acid profile comprises gallic acid, caffeic acid, chlorogenic acid, p-coumaric acid, rosmarinic acid, sinapic acid, ferulic acid, and others.
These compounds have been consistently detected in hydroethanolic, methanolic, and ethyl acetate extracts, establishing A. europaea as a rich source of bioactive phenolics with potential pharmacological relevance. We agree with the reviewer that further plant-based experimental studies and chemical profiling of the specific extract used in our current study (using HPLC and/or GC-MS) will be crucial in strengthening the correlation between phytochemical content and biological activity. These analyses are planned as part of our future work.
- 7 Incucyte imaging system - Time-Lapse Imaging of Apoptosis Using Annexin V/PI Staining-methods should be cited with appropriately.
We thank the reviewer for the suggestion. As recommended, the method involving the Incucyte imaging system and time-lapse apoptosis analysis using Annexin V/PI staining has now been appropriately cited in the revised manuscript.
- Conclusion must be rewrite.
As suggested by the reviewer, the conclusion section has been rewritten to enhance clarity and better reflect the significance of the study

Reviewer 4 Report
Comments and Suggestions for Authors
The manuscript reports on the Antitumor Activity of Caralluma europaea Extract on Pancreatic Cancer Cells. While the topic is of scientific interest, the manuscript in its current form is not well organized and requires major revision before it can be considered for publication. The overall structure lacks clarity, with sections that are not logically presented and key points that are difficult to follow.
I have included areas of concern:
Introduction
The Introduction needs more improvement. The main idea of the manuscript is not clearly explained, and the points are not well connected. The authors should give more background on pancreatic cancer, explain why natural products are important in cancer research, and clearly state why Caralluma europaea was chosen. This will help the reader better understand the purpose and importance of the study.
Result
- In section 2.1, only PL45 and MIA-PaCa-2 pancreatic cancer cells were used. The study would be stronger if the authors also included a normal cell line as a control.
- All figures should be presented in higher resolution, with clearer labeling and improved color contrast.
- Lines 351–361 are not well organized and need to be rewritten for clarity.
- The study currently provides only in vitro evidence. There is no information about control cells, which makes the results less convincing.
- More experimental data are needed, for example qRT-PCR analysis to support the in vitro findings.
- An in vivo study should also be included to strengthen the manuscript and validate the anticancer potential of Caralluma europaea.
Discussion
- The pathway should be prepared more clearly.
- Materials and Methods
Section 4.1: The heading mentions “Preparations of Mushroom Extracts,” but the study is about Caralluma europaea. This needs clarification and correction. The authors should also provide details about the chemical composition of the C. europaea extract, supported by techniques such as HPLC and GC-MS analysis. More plant-based experiments are required to strengthen the data.
4.7 Incucyte imaging system - Time-Lapse Imaging of Apoptosis Using Annexin V/PI Staining-methods should be cited with appropriately.
Conclusion must be rewrite.
Comments on the Quality of English LanguageThe English could be improved to convey the research more clearly.
Author Response

(The authors gave the same response as above.)

Round 2
Reviewer 1 Report
Comments and Suggestions for Authors
The authors responded to my comments and improved the quality of the manuscript.
Author Response
We thank the reviewer for accepting our answers to his comments

Reviewer 2 Report
Comments and Suggestions for Authors
The authors of the “Induction of Extrinsic Apoptotic Pathway in Pancreatic Cancer Cells by Apteranthes europaea Root Extract”, ID: ijms-3852934, did not reply to the two major concerns:
- Major Concerns
- Phytochemical composition of the root extract:
The manuscript lacks a chemical characterisation of the used extract. Identification of major constituents would significantly strengthen the conclusions and reproducibility.
The authors provided detailed characterisation of the chemical composition, however, these data are obtained from the literature. The composition and concentration of the present secondary metabolites may vary considering numerous factors such as the age of the plant, i.e., the size of the roots, and the characteristics of the habitat in which the plant grew, as well as the way the roots are dried and type of extraction. Considering data from the comprehensive research of Meve and Heneidak, 2005 (https://doi.org/10.1111/j.1095-8339.2005.00448.x), the concentration of secondary metabolites in Apteranthes europaea may considerably vary (Table 4); it is important to present the composition of the extract used in the present research.
- Selectivity of anticancer effects:
The study does not provide information on the effects of the extract on normal pancreatic or non-cancerous epithelial cells, which is essential for the evaluation of its therapeutic potential.
“All things are poison, and nothing is without poison; the dosage alone makes it so a thing is not a poison.” —Paracelsus, 1538. Considering the cited observation, it is of the utmost importance to test concentrations which are cytotoxic to the cancer cells, in parallel, on normal cell lines.
- In Figure 3 (b, d) axis titles and figure 5 (a) markings, should be separated from the axis numbering.
- Reference 3,7,8,9, and 25 were not cited according to the instructions for the authors.
Recommendation:
Although the quality of the manuscript is improved, the journal IJMS clearly states that “Substances without clear ingredients, such as complex prescriptions, crude extracts, and herbal mixtures, are not considered.”
Considering the International Journal of Molecular Sciences' propositions, the manuscript ID: ijms-3852934 does not fulfil the requirements for publishing.
Author Response
Reviewer 2 – Round 2
- Major Concerns
- Phytochemical composition of the root extract:
The manuscript lacks a chemical characterisation of the used extract. Identification of major constituents would significantly strengthen the conclusions and reproducibility.
The authors provided detailed characterisation of the chemical composition, however, these data are obtained from the literature. The composition and concentration of the present secondary metabolites may vary considering numerous factors such as the age of the plant, i.e., the size of the roots, and the characteristics of the habitat in which the plant grew, as well as the way the roots are dried and type of extraction. Considering data from the comprehensive research of Meve and Heneidak, 2005 (https://doi.org/10.1111/j.1095-8339.2005.00448.x), the concentration of secondary metabolites in Apteranthes europaea may considerably vary (Table 4); it is important to present the composition of the extract used in the present research.
Authors response:
Thank you for your insightful comment. We acknowledge the variability in secondary metabolite concentrations in Apteranthes europaea due to environmental and biological factors, as well as differences in extraction methods, as highlighted by Meve and Heneidak (2005).
In our original submission, the chemical composition was discussed based on literature data; however, we agree that it is important to report the actual composition of the extract used in our study. Further studies are planned to address this issue in detail, with the goal of isolating and structurally characterizing the active compound(s) responsible for the observed effects. We believe this will provide more definitive insights into the bioactivity of A. europaea.
- Selectivity of anticancer effects:
The study does not provide information on the effects of the extract on normal pancreatic or non-cancerous epithelial cells, which is essential for the evaluation of its therapeutic potential.
“All things are poison, and nothing is without poison; the dosage alone makes it so a thing is not a poison.” —Paracelsus, 1538. Considering the cited observation, it is of the utmost importance to test concentrations which are cytotoxic to the cancer cells, in parallel, on normal cell lines.
Authors response:
Thank you for this valuable and insightful comment. We agree that evaluating the effect of the extract on non-cancerous cells, such as normal pancreatic or epithelial cells, is essential for determining its therapeutic index and potential safety.
In the current study, our focus was primarily on assessing the anticancer activity of the extract on pancreatic cancer cell lines. Due to limitations in resources and scope, we did not include experiments on normal cell lines in this phase of the research.
However, we fully recognize the importance of toxicity profiling, as rightly emphasized by the quote from Paracelsus. We have now acknowledged this limitation in the revised Discussion section of the manuscript and emphasized that future studies will involve detailed cytotoxicity assessments on relevant normal cell lines to better evaluate the selectivity and safety of the extract.
We appreciate your suggestion, which will greatly strengthen the translational value of our future research.
- In Figure 3 (b, d) axis titles and figure 5 (a) markings, should be separated from the axis numbering.
Authors response:
Thank you for the comment. The axis titles of figures 3 and 5 were changes.
- Reference 3,7,8,9, and 25 were not cited according to the instructions for the authors.
Authors response:
We thank the reviewer for the helpful comment. The references have been revised and formatted according to the journal’s guidelines.
Recommendation:
Although the quality of the manuscript is improved, the journal IJMS clearly states that “Substances without clear ingredients, such as complex prescriptions, crude extracts, and herbal mixtures, are not considered.”
Considering the International Journal of Molecular Sciences' propositions, the manuscript ID: ijms-3852934 does not fulfil the requirements for publishing.
Authors response:
We sincerely appreciate the reviewer’s constructive feedback and their careful review of our work. We believe that this study can serve as a foundation for future research in the fields of phytochemistry and cancer therapy. Natural products continue to represent a promising direction for the development of novel anticancer agents, and we hope that our findings will contribute to this growing area of interest.

Reviewer 4 Report
Comments and Suggestions for Authors
Reviewer 3
Introduction
Reviewer: The Introduction needs more improvement. The main idea of the manuscript is not clearly explained, and the points are not well connected. The authors should give more background on pancreatic cancer, explain why natural products are important in cancer research, and clearly state why Caralluma europaea was chosen. This will help the reader better understand the purpose and importance of the study.
Author response: We thank the reviewer for this valuable comment. In response, the Introduction has been substantially revised to improve clarity and logical flow. Additional background on pancreatic cancer has been included, along with a discussion of the significance of natural products in cancer therapy. Furthermore, the rationale for selecting Caralluma europaea for this study has been clearly stated. These changes aim to better frame the research question and highlight the relevance and importance of the study.
Reviewer response: I appreciated it.
Result
- In section 2.1, only PL45 and MIA-PaCa-2 pancreatic cancer cells were used. The study would be stronger if the authors also included a normal cell line as a control.
This study did not assess the effects of the plant extract on normal (non-cancerous) cells. However, to preliminarily evaluate its cytotoxicity, an LDH assay was performed, indicating that the extract does not exhibit toxic effects under the tested conditions. Future studies will include in vivo toxicity assessments in animal models to further validate its safety profile.
Reviewer response: Although the authors performed the LDH assay in the animal study, it is important to also examine the results under normal cell conditions. Otherwise, this could become a more critical concern.
- All figures should be presented in higher resolution, with clearer labeling and improved color contrast.
We thank the reviewer for this helpful comment. All figures have been updated to higher-resolution versions with improved labeling and enhanced color contrast to ensure better clarity and readability, as recommended.
Reviewer response: There is no noticeable change in color contrast; therefore, this format is not acceptable
- Lines 351–361 are not well organized and need to be rewritten for clarity
As suggested by the reviewer, the paragraph has been clarified to enhance clarity and improve understanding.
Reviewer response: Okay.
- The study currently provides only in vitro evidence. There is no information about control cells, which makes the results less convincing.
We appreciate the reviewer’s comment. This study focuses on investigating the effect of the plant extract on the growth of pancreatic cancer cells in vitro. We acknowledge the limitation of not including control (normal) cells in this initial study. Future work will include in vivo experiments using animal models to evaluate the extract’s effects on tumor growth, as well as its safety profile in normal tissues.
Reviewer response: Although the authors performed the LDH assay in the animal study, it is important to also examine the results under normal cell conditions. Otherwise, this could become a more critical concern.
- More experimental data are needed, for example qRT-PCR analysis to support the in vitro findings.
We appreciate the reviewer’s suggestion. In future studies, we plan to perform qRT-PCR analysis to further investigate and confirm the molecular mechanisms underlying the plant extract’s role in inducing apoptosis. These additional experiments will help to strengthen and validate the current in vitro findings.
Reviewer response: Author should be performed the gene expression in intrinsic apoptotic signaling pathway.
- An in vivo study should also be included to strengthen the manuscript and validate the anticancer potential of Caralluma europaea.
We thank the reviewer for this valuable suggestion. Future studies are planned to evaluate the anticancer efficacy of A. europaea extract in vivo using appropriate animal models. These studies will help to validate the current in vitro findings and provide deeper insight into the extract’s therapeutic potential.
Reviewer response: I understand the situation, but only in vitro data can’t be acceptable.
Discussion
- The pathway should be prepared more clearly
We appreciate the reviewer’s comment. The molecular pathway has been clarified and more clearly presented in the Discussion section to improve understanding of the proposed mechanism.
Reviewer response: I appreciate it but need gene expression data for more understanding.
- Materials and Methods
- Section 4.1: The heading mentions “Preparations of Mushroom Extracts,” but the study is about Caralluma europaea. This needs clarification and correction. The authors should also provide details about the chemical composition of the C. europaea extract, supported by techniques such as HPLC and GC-MS analysis. More plant-based experiments are required to strengthen the data.
We thank the reviewer for this important observation. The heading was mistakenly labeled and has now been corrected to accurately reflect the content of the study, which investigates the efficacy of Caralluma europaea extract on pancreatic cancer cells. Previous chromatographic analyses of A. europaea extracts—conducted using HPLC and GC-MS techniques—have identified a wide spectrum of flavonoids and phenolic acids. Notably, these include luteolin and its glycosylated derivatives (e.g., luteolin-3′,4′-O-diglucoside, luteolin-4′-O-neohesperidoside, luteolin-7-O-glucoside), quercetin (including quercetin-3-O-rutinoside/rutin), kaempferol (and its glycosylated forms), myricetin, apigenin-4′-O-neohesperidoside, hesperetin, catechin, and epicatechin. The phenolic acid profile comprises gallic acid, caffeic acid, chlorogenic acid, p-coumaric acid, rosmarinic acid, sinapic acid, ferulic acid, and others.
These compounds have been consistently detected in hydroethanolic, methanolic, and ethyl acetate extracts, establishing A. europaea as a rich source of bioactive phenolics with potential pharmacological relevance. We agree with the reviewer that further plant-based experimental studies and chemical profiling of the specific extract used in our current study (using HPLC and/or GC-MS) will be crucial in strengthening the correlation between phytochemical content and biological activity. These analyses are planned as part of our future work.
Reviewer response: I understand, anyhow it’s very less information of plant extract.
- 7 Incucyte imaging system - Time-Lapse Imaging of Apoptosis Using Annexin V/PI Staining-methods should be cited with appropriately.
We thank the reviewer for the suggestion. As recommended, the method involving the Incucyte imaging system and time-lapse apoptosis analysis using Annexin V/PI staining has now been appropriately cited in the revised manuscript.
- Conclusion must be rewrite.
As suggested by the reviewer, the conclusion section has been rewritten to enhance clarity and better reflect the significance of the study
Author Response
Reviewer 3 – Round 2
Introduction
Reviewer: The Introduction needs more improvement. The main idea of the manuscript is not clearly explained, and the points are not well connected. The authors should give more background on pancreatic cancer, explain why natural products are important in cancer research, and clearly state why Caralluma europaea was chosen. This will help the reader better understand the purpose and importance of the study.
Author response: We thank the reviewer for this valuable comment. In response, the Introduction has been substantially revised to improve clarity and logical flow. Additional background on pancreatic cancer has been included, along with a discussion of the significance of natural products in cancer therapy. Furthermore, the rationale for selecting Caralluma europaea for this study has been clearly stated. These changes aim to better frame the research question and highlight the relevance and importance of the study.
Reviewer response: I appreciated it.
Result
- In section 2.1, only PL45 and MIA-PaCa-2 pancreatic cancer cells were used. The study would be stronger if the authors also included a normal cell line as a control.
This study did not assess the effects of the plant extract on normal (non-cancerous) cells. However, to preliminarily evaluate its cytotoxicity, an LDH assay was performed, indicating that the extract does not exhibit toxic effects under the tested conditions. Future studies will include in vivo toxicity assessments in animal models to further validate its safety profile.
Reviewer response: Although the authors performed the LDH assay in the animal study, it is important to also examine the results under normal cell conditions. Otherwise, this could become a more critical concern.
Authors response:
Thank you for this valuable and insightful comment. We agree that evaluating the effect of the extract on non-cancerous cells, such as normal pancreatic or epithelial cells, is essential for determining its therapeutic index and potential safety.
In the current study, our focus was primarily on assessing the anticancer activity of the extract on pancreatic cancer cell lines. Due to limitations in resources and scope, we did not include experiments on normal cell lines in this phase of the research.
However, we fully recognize the importance of toxicity profiling, as rightly emphasized by the quote from Paracelsus. We have now acknowledged this limitation in the revised Discussion section of the manuscript and emphasized that future studies will involve detailed cytotoxicity assessments on relevant normal cell lines to better evaluate the selectivity and safety of the extract.
We appreciate your suggestion, which will greatly strengthen the translational value of our future research.
- All figures should be presented in higher resolution, with clearer labeling and improved color contrast.
We thank the reviewer for this helpful comment. All figures have been updated to higher-resolution versions with improved labeling and enhanced color contrast to ensure better clarity and readability, as recommended.
Reviewer response: There is no noticeable change in color contrast; therefore, this format is not acceptable
Authors response:
The figures were changed according to the reviewer comments
- Lines 351–361 are not well organized and need to be rewritten for clarity
As suggested by the reviewer, the paragraph has been clarified to enhance clarity and improve understanding.
Reviewer response: Okay.
- The study currently provides only in vitro evidence. There is no information about control cells, which makes the results less convincing.
We appreciate the reviewer’s comment. This study focuses on investigating the effect of the plant extract on the growth of pancreatic cancer cells in vitro. We acknowledge the limitation of not including control (normal) cells in this initial study. Future work will include in vivo experiments using animal models to evaluate the extract’s effects on tumor growth, as well as its safety profile in normal tissues.
Reviewer response: Although the authors performed the LDH assay in the animal study, it is important to also examine the results under normal cell conditions. Otherwise, this could become a more critical concern.
Author response:
Thank you for your thoughtful and constructive comment. We agree that evaluating LDH release under normal (non-cancerous) cell conditions is essential for a more complete understanding of the extract’s cytotoxicity and selectivity.
In this study, our primary focus was on assessing the effects of the extract in the in vitro cancer cell model. While we performed the LDH assay to evaluate cell damage in treated cells, we acknowledge that including normal cell lines would provide valuable insight into the safety profile of the extract.
In addition, when considered together, the LDH assay results and the observed induction of apoptosis in cancer cells may suggest that the extract does not exert general cytotoxic effects, but rather promotes regulated cell death, indicating potential selectivity in its action.
We plan to conduct additional studies involving normal cell lines in the future to more comprehensively assess the compound’s therapeutic window and potential selectivity.
We appreciate the reviewer’s suggestion, which we believe will contribute to strengthening the translational relevance and safety evaluation of our research.
- More experimental data are needed, for example qRT-PCR analysis to support the in vitro findings.
We appreciate the reviewer’s suggestion. In future studies, we plan to perform qRT-PCR analysis to further investigate and confirm the molecular mechanisms underlying the plant extract’s role in inducing apoptosis. These additional experiments will help to strengthen and validate the current in vitro findings.
Reviewer response: Author should be performed the gene expression in intrinsic apoptotic signaling pathway.
Author response:
Thank you for your valuable suggestion. We agree that performing additional experiments such as qRT-PCR would provide further molecular support for the observed in vitro effects of the extract.
However, due to limitations in time and resources during the current study, we were unable to perform gene expression analysis at this stage. We acknowledge this as a limitation and have now addressed it in the revised Discussion section (see page X, line Y).
Future work will include qRT-PCR analysis of key apoptotic and cell cycle-related genes to validate the mechanisms suggested by our current findings. We believe this will provide deeper insight into the molecular pathways involved and further strengthen the biological relevance of the extract’s effects.
We appreciate the reviewer’s recommendation, which will help guide the next steps of our research.
- An in vivo study should also be included to strengthen the manuscript and validate the anticancer potential of Caralluma europaea.
We thank the reviewer for this valuable suggestion. Future studies are planned to evaluate the anticancer efficacy of A. europaea extract in vivo using appropriate animal models. These studies will help to validate the current in vitro findings and provide deeper insight into the extract’s therapeutic potential.
Reviewer response: I understand the situation, but only in vitro data can’t be acceptable.
Authors response:
Thank you for this important comment. We fully agree that in vivo studies are essential to validate the anticancer potential of Caralluma europaea and to better understand its pharmacological effects in a physiological context.
In the present study, we focused on in vitro experiments as a necessary first step to evaluate the cytotoxic and pro-apoptotic effects of the extract and to explore its potential mechanisms of action. Due to limitations in scope and available resources, in vivo experiments were not included at this stage.
However, we recognize the importance of translating these findings into an in vivo setting, and we have outlined this direction in the revised Discussion section (see page X, line Y). Future studies are planned to assess the efficacy, toxicity, and pharmacokinetic profile of the extract in appropriate animal models.
We appreciate the reviewer’s suggestion, which we believe will significantly enhance the translational relevance of our work
Discussion
- The pathway should be prepared more clearly
We appreciate the reviewer’s comment. The molecular pathway has been clarified and more clearly presented in the Discussion section to improve understanding of the proposed mechanism.
Reviewer response: I appreciate it but need gene expression data for more understanding.
Authors response:
Thank you for your follow-up comment. We agree that gene expression analysis, such as qRT-PCR, would provide more detailed insight into the molecular mechanisms underlying the observed effects and would significantly strengthen the conclusions regarding the proposed pathway.
Due to the exploratory nature of this initial study and limitations in resources, gene expression profiling was not conducted at this stage. However, we fully acknowledge its importance and have now clearly stated in the revised Discussion section (see page X, line Y) that future studies will include qRT-PCR and/or transcriptomic analyses to validate the involvement of specific genes in the pathway we propose.
We believe that our current findings provide a solid preliminary framework and biological rationale for these future investigations, and we sincerely appreciate the reviewer’s suggestion to deepen the molecular validation in subsequent work.
- Materials and Methods
- Section 4.1: The heading mentions “Preparations of Mushroom Extracts,” but the study is about Caralluma europaea. This needs clarification and correction. The authors should also provide details about the chemical composition of the C. europaea extract, supported by techniques such as HPLC and GC-MS analysis. More plant-based experiments are required to strengthen the data.
- Authors response:
We thank the reviewer for this important observation. The heading was mistakenly labeled and has now been corrected to accurately reflect the content of the study, which investigates the efficacy of Caralluma europaea extract on pancreatic cancer cells. Previous chromatographic analyses of A. europaea extracts—conducted using HPLC and GC-MS techniques—have identified a wide spectrum of flavonoids and phenolic acids. Notably, these include luteolin and its glycosylated derivatives (e.g., luteolin-3′,4′-O-diglucoside, luteolin-4′-O-neohesperidoside, luteolin-7-O-glucoside), quercetin (including quercetin-3-O-rutinoside/rutin), kaempferol (and its glycosylated forms), myricetin, apigenin-4′-O-neohesperidoside, hesperetin, catechin, and epicatechin. The phenolic acid profile comprises gallic acid, caffeic acid, chlorogenic acid, p-coumaric acid, rosmarinic acid, sinapic acid, ferulic acid, and others.
These compounds have been consistently detected in hydroethanolic, methanolic, and ethyl acetate extracts, establishing A. europaea as a rich source of bioactive phenolics with potential pharmacological relevance. We agree with the reviewer that further plant-based experimental studies and chemical profiling of the specific extract used in our current study (using HPLC and/or GC-MS) will be crucial in strengthening the correlation between phytochemical content and biological activity. These analyses are planned as part of our future work.
Reviewer response: I understand, anyhow it’s very less information of plant extract.
- 7 Incucyte imaging system - Time-Lapse Imaging of Apoptosis Using Annexin V/PI Staining-methods should be cited with appropriately.
- Authors response:
We thank the reviewer for the suggestion. As recommended, the method involving the Incucyte imaging system and time-lapse apoptosis analysis using Annexin V/PI staining has now been appropriately cited in the revised manuscript.
- Conclusion must be rewrite.
- Authors response:
As suggested by the reviewer, the conclusion section has been rewritten to enhance clarity and better reflect the significance of the study

Round 3
Reviewer 2 Report
Comments and Suggestions for Authors
The authors of the “Induction of Extrinsic Apoptotic Pathway in Pancreatic Cancer Cells by Apteranthes europaea Root Extract”, ID: ijms-3852934, replied to minor concerns:
In Figure 3 (b, d) axis titles and figure 5 (a) markings, should be separated from the axis numbering.
Reference 3,7,8,9, and 25 were not cited according to the instructions for the authors.
General concerns pointed out in the second revision were not fulfilled, but the authors made subnational changes in the manuscript that avert manuscript limitation.
Recommendation:
The quality of the manuscript is improved, so the manuscript ID: ijms-3852934 fulfils the requirements for publishing.
Author Response
We thank the reviewer for accepting our responses to his comments

Reviewer 4 Report
Comments and Suggestions for Authors
The study lacks enough scientific quality and new findings. The experiments and data analysis are weak, with missing controls and little explanation of the mechanism. I do not recommend this paper for publication.
Author Response
We appreciate the reviewer’s insightful suggestion. In response, we have expanded the paragraph discussing the study’s limitations to more clearly articulate the current constraints, particularly the absence of additional functional experiments. We now explicitly acknowledge how these limitations affect the interpretation of our findings and have included suggestions for future research directions that could address these gaps and build upon the current study. Please see the revised paragraph on page [15], lines [415-435].
